# Discovery of fungal onoceroid triterpenoids through domainless enzyme-targeted global genome mining

Jia Tang [1] & Yudai Matsuda [1] ✉

Genomics-guided methodologies have revolutionized the discovery of natural products. However, a major challenge in the field of genome mining is determining how to selectively extract biosynthetic gene clusters (BGCs) for untapped natural products from numerous available genome sequences. In this study, we developed a fungal genome mining tool that extracts BGCs encoding enzymes that lack a detectable protein domain (i.e., domainless enzymes) and are not recognized as biosynthetic proteins by existing bioinformatic tools. We searched for BGCs encoding a homologue of Pyr4-family terpene cyclases, which are representative examples of apparently domainless enzymes, in approximately 2000 fungal genomes and discovered several BGCs with unique features. The subsequent characterization of selected BGCs led to the discovery of fungal onoceroid triterpenoids and unprecedented onoceroid synthases. Furthermore, in addition to the onoceroids, a previously unreported sesquiterpene hydroquinone, of which the biosynthesis involves a Pyr4-family terpene cyclase, was obtained. Our genome mining tool has broad applicability in fungal genome mining and can serve as a beneficial platform for accessing diverse, unexploited natural products.

Recent years have witnessed an exponential increase in data accumulation across all fields, ushering us into the era of big data. Research on naturally occurring organic compounds, typically referred to as natural products, is no exception to this global trend[1,2]. Traditional methods used to isolate natural products often lead to the rediscovery of known compounds, and therefore, scientists are now required to discover hidden natural products or synthesize their analogues using alternative strategies. Leveraging big data presents a promising solution to this problem. The rapid accumulation of microbial genome sequence data has revealed that microbial genomes harbor more natural product biosynthetic gene clusters (BGCs) than one can predict based on the number of metabolites produced under standard laboratory cultivation conditions[3–5]. Thus, the activation and characterization of these silent or orphan BGCs present in microbial genomes can lead to the discovery of previously undescribed natural products. A method known as genome mining has been demonstrated as useful for discovering unexploited metabolites in recent decades[6–8].

However, a major challenge in the field of genome mining is determining how to prioritize BGCs identified in available genome sequence data for the efficient discovery of untapped natural products.

Fungi are prolific producers of natural products with diverse molecular structures and biological activities, establishing them as promising sources of unexplored metabolites. To facilitate genome mining in fungi, several bioinformatic tools, such as antiSMASH[9], SMURF[10], DeepBGC[11], and TOUCAN[12], can be employed to detect BGCs in a specific fungal genome. Among the available tools, antiSMASH has gained the most popularity, possibly because of its user-friendly online platform and versatile functions. However, some challenges have been encountered while using antiSMASH for BGC detection and genome mining. First, antiSMASH classifies many genes as other genes, even when they encode a biosynthetic enzyme. For example, when antiSMASH (version 7.0.0) was employed to analyze the BGC of the fungal meroterpenoid novofumigatonin[13], a hybrid molecule of polyketide and terpenoid origin, it categorized 6 out of 13 genes as other genes.

[1]Department of Chemistry, City University of Hong Kong, Tat Chee Avenue, Kowloon, Hong Kong SAR, China. ✉e-mail: ymatsuda@cityu.edu.hk

This may be because some of these genes lacked a detectable Pfam domain[14] (Supplementary Fig. 1). Moreover, one of these six overlooked genes encodes the terpene cyclase NvfL, which has widely distributed homologues in natural product pathways[15]. Thus, although antiSMASH should theoretically categorize this BGC as "T1PKS, terpene" (type I polyketide synthase + terpene synthase), it recognized only T1PKS. In addition, these enzymes without a detectable protein domain (hereafter termed domainless enzymes) included the α-ketoglutarate-dependent dioxygenase NvfI and the methyltransferase NvfJ, indicating that antiSMASH failed to recognize various biosynthetic proteins. Furthermore, although antiSMASH and the other aforementioned tools can be used to extract all possible BGCs from a genome sequence, they do not facilitate the selective extraction of BGCs with a desired or specific feature. Thus, users need to select BGCs based on the criteria set for subsequent wet lab experiments. In this study, we aimed to overcome these challenges by developing a widely applicable genome mining methodology for the rapid discovery of unexplored natural products and biosynthetic mechanisms.

## Results

### Development of a fungal genome mining tool

Recent studies on the biosynthesis of fungal natural products have identified many biosynthetic enzymes lacking a detectable Pfam domain, such as NvfI and NvfL, which are not recognized by antiSMASH. Because these domainless biosynthetic enzymes occasionally drive intriguing chemical reactions, we hypothesized that a genome mining strategy focusing on BGCs encoding weak homologues of such enzymes would offer a promising approach to obtaining untapped metabolites. To identify biosynthetic proteins without detectable Pfam domains, we first collected known fungal biosynthetic proteins. Although these proteins can be obtained from the UniProtKB/Swiss-Prot or MIBiG database[16], these databases contain gene or protein sequences that can be mispredicted and are not manually corrected. Thus, we created our own fungal BGC database by manually collecting, reviewing, correcting where necessary, and systematically annotating approximately 700 fungal BGCs (Supplementary Data 1). We named this database the FunBGCs database.

Next, we extracted all protein sequences from the created database and conducted a Pfam domain search on the resulting 5070 proteins. This led to the identification of 520 Pfam domains. However, 572 proteins lacked a detectable Pfam domain. These domainless proteins were divided into groups based on their sequence similarity,

and their hidden Markov model (HMM) profiles were generated for HMMER[17] analysis. The representative members of these protein groups include the terpene cyclase Pyr4[18], the Diels−Alderase Fsa2[19], the hetero Diels−Alderase AsR5[20], the epoxide hydrolase CtvD[21], and the isomerase Trt14[22]. Furthermore, several additional HMM profiles were created for the domains of polyketide synthases (PKSs) and nonribosomal peptide synthetases (NRPSs). The identified Pfam domains, the manually added protein families or domains, and a few additional protein domains from the SMART[23] and TIGRFAMs[24] databases (Supplementary Data 2 and 3) were combined for use in biosynthetic protein detection. Moreover, a DIAMOND[25] database was constructed with all of the extracted biosynthetic proteins.

BGCs were extracted as follows (Fig. 1a). Initially, genes whose products belonged to a protein family of interest were identified from a given fungal genome. Subsequently, genes in the flanking regions of each identified gene were examined to determine if they encoded a core protein (e.g., PKSs, NRPSs, terpene synthases or cyclases, and prenyltransferases; refer to Supplementary Table 1 for more details) or another type of biosynthetic protein using the previously created in-house HMM library and DIAMOND database. If a target gene was found to be colocalized with a core protein gene, then the genomic region was extracted as a BGC by applying a similar strategy used for other fungal genome mining tools, such as antiSMASH[9] and SMURF[10]. To determine whether a gene in flanking regions was clustered with the target, the following factors were considered: (i) whether the gene encoded a biosynthetic protein, (ii) whether the gene was duplicated in the genome (to include a possible self-resistance enzyme gene[26]), (iii) whether the gene was biosynthetically related to nearby genes, (iv) whether the gene encoded a small protein (to ignore mispredicted small proteins), and (v) the distance between a given gene and the next one (refer to the Methods section for the detailed procedure). In addition, the extracted BGCs were visualized using a web browser, which displayed general information on each BGC and each gene (Fig. 1b). The fungal genome mining tool developed in this study was named FunBGCeX (Fungal BGC eXtractor).

### Discovery of fungal BGCs with unusual features

To examine whether our genome mining tool allows for the efficient discovery of BGCs that potentially synthesize an unreported class of natural products, we extracted BGCs that encoded a homologue of Pyr4 (Fig. 2a)[18,27,28], which is a noncanonical transmembrane terpene cyclase and does not have a conserved domain, as inferred from the

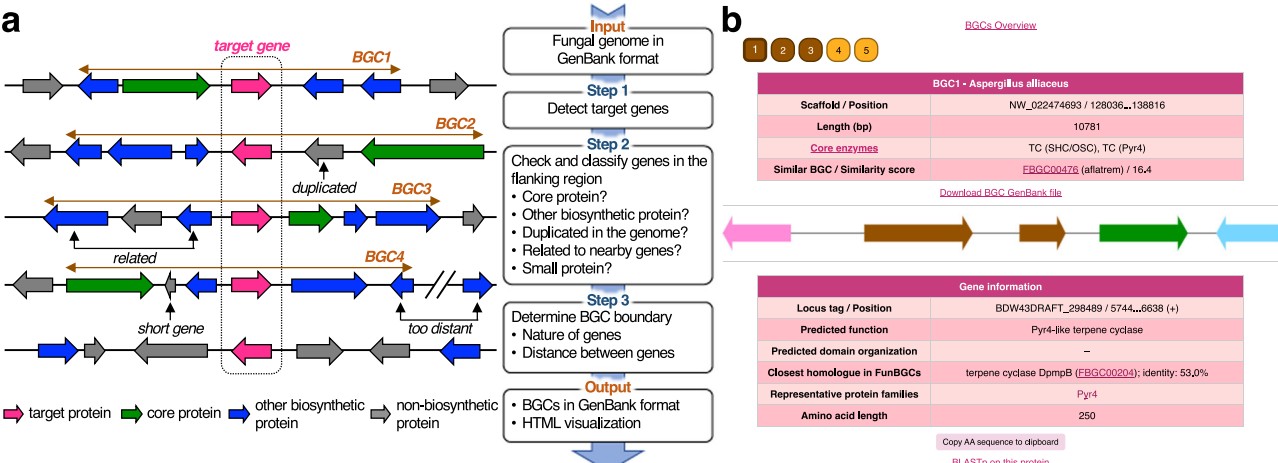

**Fig. 1 | Fungal genome mining tool. a** General workflow to extract biosynthetic gene clusters (BGCs) with a target gene from a given fungal genome. **b** Example of output from the genome mining tool. For this panel, BGCs encoding a Pyr4 homologue were extracted from the fungus *Aspergillus alliaceus* CBS 536.65, yielding five BGCs. The top table provides general information on a BGC. A schematic representation of BGC is displayed in the middle. On clicking each gene, information on that gene is provided in the bottom table. The BGC shown here is identical to the *alli* cluster (Fig. 3a).

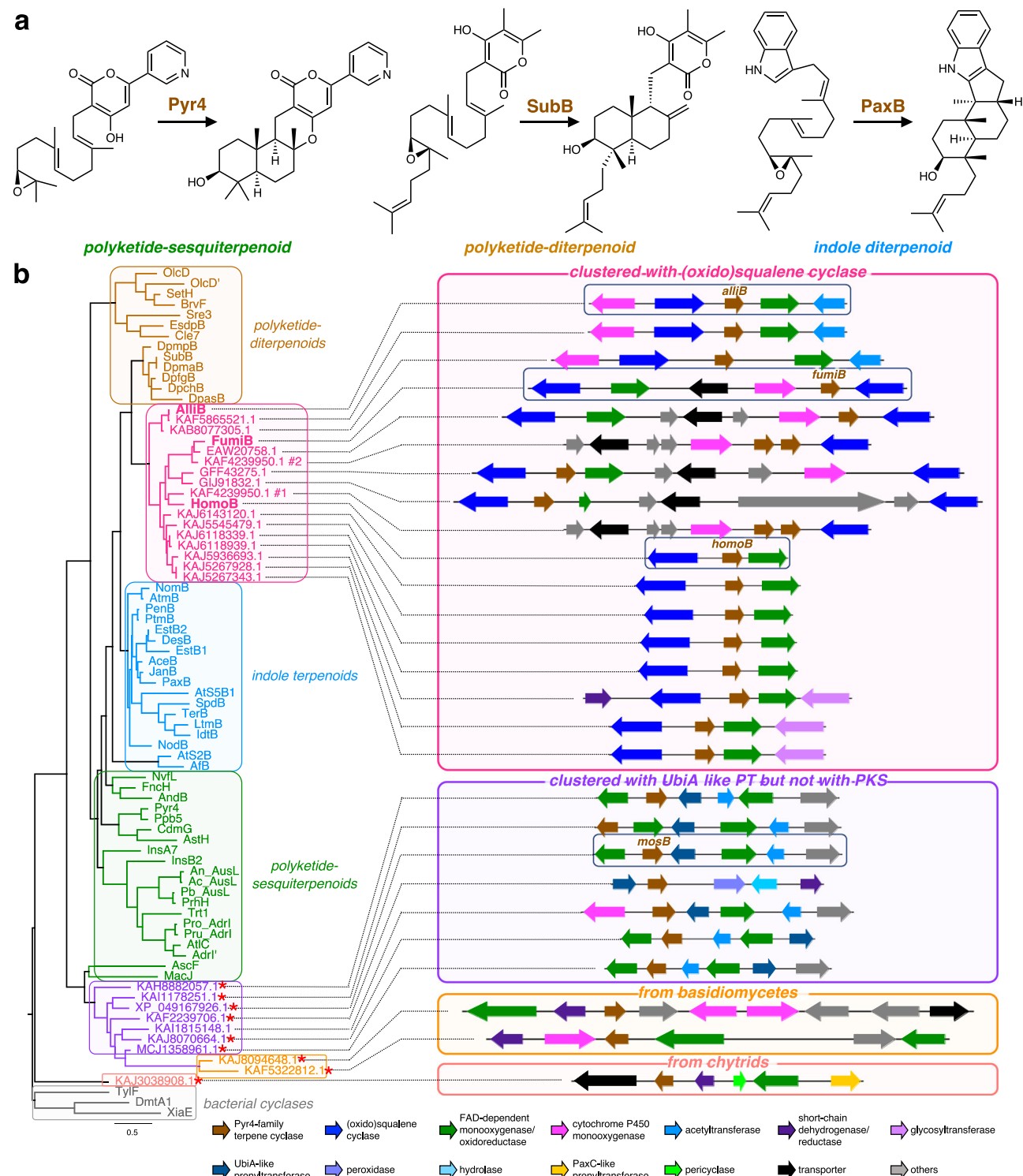

**Fig. 2 | Discovery of atypical BGCs encoding a Pyr4-family terpene cyclase.**
**a** Reactions catalyzed by the selected members of Pyr4-family terpene cyclases.
**b** Phylogenetic analysis of known Pyr4 homologues and those identified in this study, along with their associated biosynthetic gene clusters. Red asterisks indicate

proteins not detected by antiSMASH analysis. The gene and protein sequences of several identified Pyr4 homologues were manually revised upon creating the phylogenetic tree (refer to Supplementary Data 7 for their protein sequences).

National Center for Biotechnology Information (NCBI) Conserved Domain Search[29]. Thus, such BGCs were extracted from 1990 annotated fungal reference genomes downloaded from the NCBI database (Supplementary Data 4) using an HMM profile created from Pyr4 and its homologues[15,30]. This process resulted in the identification of 1050 BGCs (including those with a single gene; Supplementary Data 5). More

than half (690) of the identified BGCs did not contain an additional core protein gene, whereas 182 BGCs possessed at least one PKS gene, and 131 BGCs were found to encode a PaxC-like prenyltransferase, which is specifically required for the biosynthesis of indole sesqui-terpenoids and diterpenoids[31]. Thus, except for standalone *pyr4*-like genes, the majority of the *pyr4*-like genes were clustered with either a

PKS gene or a *paxC*-like gene. This observation was consistent with the fact that, in fungi, Pyr4-family terpene cyclases have been found only in the biosynthetic pathways of polyketide–terpenoid hybrids and indole terpenoids[32,33].

We then focused on BGCs that show low similarity with known fungal BGCs and encode a Pyr4-like protein as well as a core enzyme different from PKS or PaxC-like prenyltransferase, with the expectation that such BGCs are responsible for the synthesis of unusual metabolites. After manual inspection of the raw output from the genome mining tool, we found that 16 BGCs encoded one or two protein(s) homologous to squalene–hopene cyclases (SHCs) or oxidosqualene cyclases (OSCs), which are involved in the biosynthesis of triterpenoids. Intriguingly, Pyr4-like proteins encoded by these BGCs formed a distinct clade when subjected to phylogenetic analysis with known Pyr4-family terpene cyclases (Fig. 2b). During our study, we also observed that the genome of *Aspergillus fumigatus* A1163, a non-reference genome of *A. fumigatus*, contained a similar BGC, which was reserved for further investigation. To date, no metabolic pathway involving both a Pyr4-family terpene cyclase and an SHC/OSC-like enzyme has been identified. Furthermore, the nature of these BGCs suggests that their products are triterpenoids whose biosynthesis requires two cyclization events. However, to the best of our knowledge, none of the known fungal metabolites appear to be synthesized by these BGCs. Therefore, we speculate that these BGCs are responsible for the biosynthesis of untapped triterpenoid molecules. In addition, we found seven BGCs that encoded a UbiA-like prenyltransferase but lacked a PKS gene (Fig. 2b). Thus, it is predicted that these UbiA-like prenyltransferases accept a non-polyketide substrate to attach a prenyl group, which would then be cyclized by the Pyr4-family terpene cyclase encoded by these BGCs. However, such a biosynthetic mechanism has never been reported, and therefore, these BGCs might be involved in the formation of meroterpenoids with a non-polyketide moiety.

In terms of the sources of the Pyr4-like proteins identified in this study, most of them (1010/1050) are of ascomycete origin, which is consistent with the fact that all of the characterized fungal Pyr4-family terpene cyclases are from ascomycete fungi. However, we also noted the presence of Pyr4 homologues from fungi in other divisions, namely Chytridiomycota (12/1050), Mucoromycota (3/1050), and Basidiomycota (25/1050), some of which are clustered with other potential biosynthetic genes (Fig. 2b). Two basidiomycete fungi, *Marasmius tenuissimus* GH-37 and *Psilocybe* cf. *subviscida* CBS 101986, possess homologous BGCs with a *pyr4*-like gene. These two BGCs apparently lack a prenyltransferase gene or other well-known core enzyme genes, and therefore, the Pyr4 homologues encoded by these BGCs might directly accept a ubiquitously present polyprenyl molecule. Furthermore, the chytrid fungus *Rhizophlyctis rosea* JEL0764 harbors a BGC that encodes a Pyr4-like protein and a PaxC-like prenyltransferase. Since this Pyr4 homologue is distantly related to any of the known Pyr4-family terpene cyclases, the BGC could potentially synthesize an unprecedented meroterpenoid species. Collectively, our global genome mining analysis could readily identify several different groups of unexploited BGCs in diverse fungi, which are worth studying further.

To compare the performance of our genome mining methodology with that of antiSMASH (fungiSMASH) analysis, the same set of 1990 fungal genomes used in our genome mining were analyzed by the standalone version of antiSMASH. The detected BGCs were then examined to see whether they encoded a Pyr4 homologue, resulting in the extraction of 325 BGCs (Supplementary Data 6). Although these BGCs include those encoding an (oxido)squalene cyclase, most of the other uncharacterized BGCs mentioned above could not be detected by antiSMASH. Furthermore, antiSMASH analysis is unable to tell which BGCs encode a Pyr4-like protein (or another protein of the user's interest), and therefore, the extraction of BGCs with a *pyr4* homologue by antiSMASH requires (i) detection of all BGCs from given genome

sequences and (ii) manual inspection to determine whether each BGC encodes a Pyr4 homologue. Altogether, the antiSMASH analysis fails to detect certain types of BGCs, which can be detected by our approach, and also requires a much longer time for the completion of the analysis. On the contrary, our genome mining method allows for the automated extraction of BGCs of the user's interest within a short period of time.

## Characterization of selected triterpenoid BGCs

To examine whether the unusual BGCs identified by our global genome mining analysis can yield untapped natural products, we then focused on those encoding an SHC/OSC-like protein and selected three BGCs for experimental investigation. The three BGCs were from *Aspergillus homomorphus* CBS 101889, *A. fumigatus* A1163 (CBS 144.89), and *Aspergillus alliaceus* CBS 536.65, which were designated as the *homo*, *fumi*, and *alli* clusters, respectively (Fig. 3a, Supplementary Table 2, and Supplementary Data 8). The *homo* cluster was the simplest of the three and encoded the squalene or oxidosqualene cyclase HomoS, the Pyr4-family terpene cyclase HomoB, and the FAD-dependent monooxygenase (FMO) HomoM, the last of which is homologous to epoxidases involved in fungal meroterpenoid pathways[32,33]. In addition, the homologues of the three enzymes were conserved in the other two BGCs, although the *fumi* cluster contained two *homoS* homologues, *fumiS1* and *fumiS2*. On the basis of the predicted functions of Homo enzymes, the product of the *homo* cluster could be synthesized as follows. First, HomoS cyclizes squalene or oxidosqualene to yield a cyclized product. Then, HomoM epoxidizes one of the olefinic double bonds present in the cyclized product, and HomoB finally performs a second round of cyclization to produce the end pathway product. The other two BGCs would employ a similar biosynthetic mechanism; however, they each encoded one or two tailoring enzymes predicted to modify the triterpenoid scaffold. The *fumi* cluster contained a cytochrome P450 monooxygenase gene *fumiP*, whereas the *alli* cluster encoded the P450 AlliP and the acetyltransferase AlliA. Collectively, all of the three BGCs were expected to yield different triterpenoid species.

To analyze the functions of the three BGCs and identify the metabolites produced, we performed heterologous expression experiments in *Aspergillus oryzae* NSAR1[34], which has been widely utilized for the characterization of orphan BGCs[27,35–37]. We first individually expressed the four putative triterpene synthase genes (*homoS*, *fumiS1*, *fumiS2*, and *alliS*) in *A. oryzae* and then analyzed metabolites from the resulting transformants using gas chromatography–mass spectrometry (GC–MS). The findings revealed that all of the enzymes, except FumiS2, yielded specific metabolite(s) (Fig. 3b, traces i to v). The *homoS*-transformed strain produced a major product **1** with $m/z$ 410 [M]$^+$, which was not present in the host strain. After isolation, metabolite **1** was identified as bicyclic triterpene α-polypodatetraene based on the comparison of its nuclear magnetic resonance (NMR) spectra and specific rotation with reported data[38] (Fig. 3c). Compound **1** was also obtained from the *A. oryzae* transformant harboring *fumiS1*; however, the transformant produced another product **2**, which was identified to be α-polypodatetraen-3β-ol[39], the C-3 hydroxy analogue of **1** (Fig. 3c). The metabolic profile of the *A. oryzae* strain with *alliS* differed from those of the other two. A specific metabolite **3** was detected as a major peak and was identified as 8α-hydroxypolypoda-13,17,21-triene[40] (Fig. 3c).

We then identified the downstream metabolites of the three pathways and first focused on the end product of the *homo* pathway. The co-expression of the epoxidase gene *homoM* and the *pyr4* homologue *homoB* with *homoS* resulted in the formation of an additional metabolite **4** with $m/z$ 426 [M]$^+$ (Fig. 3b, trace vi). Analysis of the NMR data of **4** revealed the presence of two additional ring systems, whose absolute structure was established using the modified Mosher's method[41] (Fig. 3c and Supplementary Fig. 2). Thus, compound **4**,

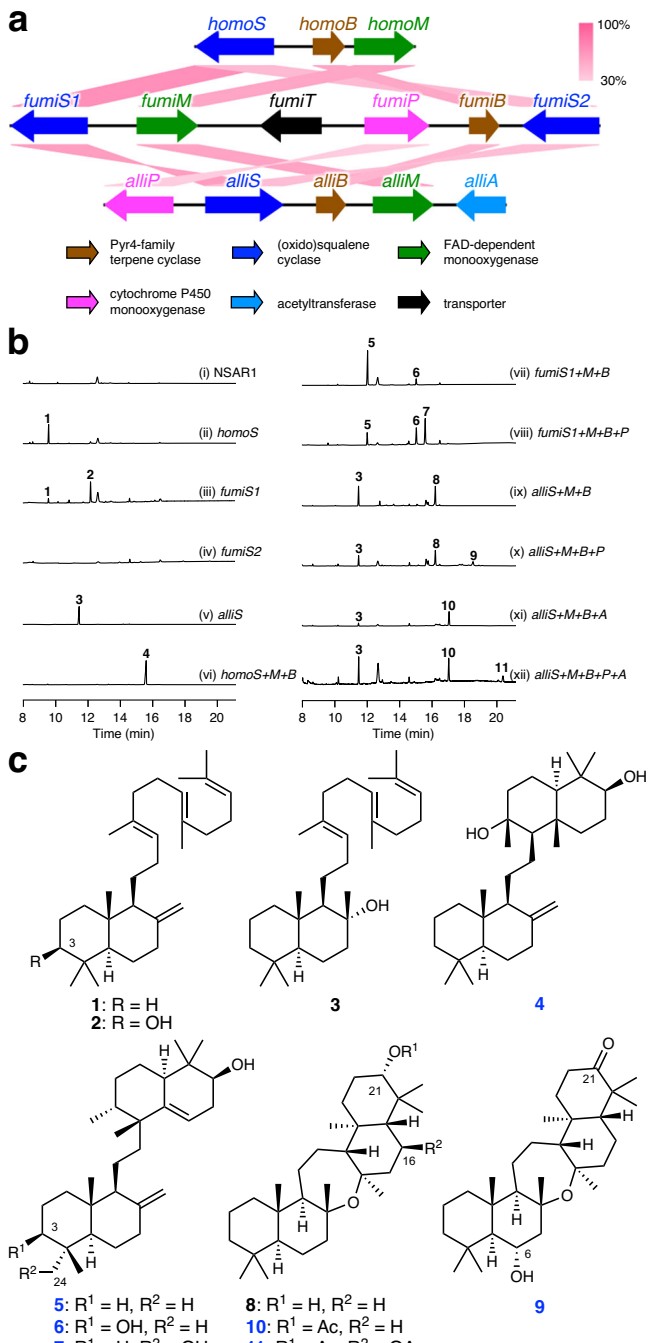

**Fig. 3 | Functional analysis of fungal onoceroid BGCs. a** Schematic representations of fungal onoceroid biosynthetic gene clusters and BLASTp comparisons of each gene product. **b** Gas chromatography–mass spectrometry profiles of the metabolites obtained from *Aspergillus oryzae* transformants. **c** Structures of triterpenoids obtained in this study. The numbers of previously undescribed compounds are given in *blue* (Note that **8** had only been obtained as a synthetic compound[42]).

hereby named homomonoceroid A, is synthesized through the cyclization of squalene from both termini and is classified as an onoceroid.

Next, the co-expression system of *fumiS1*, *fumiM*, and *fumiB* was constructed, which yielded two additional metabolites, **5** and **6** (Fig. 3b, trace vii). The major product **5**, designated as fumionoceroid A, was determined to be another onoceroid; its structure was established through NMR and X-ray crystallographic analyses (Fig. 3c and Supplementary Fig. 3; CCDC: 2279439). Compound **5** was also found to

be a tetracyclic onoceroid as **4**, and both **4** and **5** appeared to be derived from **1**. However, the newly formed bicyclic system of **5** differed from that of **4**. The other product **6**, fumionoceroid B, was determined to be the C-3 hydroxy form of **5** and appeared to be synthesized in the same manner as **5**, using oxidosqualene as a starting material (Fig. 3c). The absolute structure of **6** was deduced based on that of **5**. Subsequently, the P450 gene *fumiP* was introduced, yielding an additional product **7** (Fig. 3b, trace viii). NMR and X-ray crystallographic analyses revealed that **7**, named fumionoceroid C, was a hydroxylated form of **5** at C-24 (Fig. 3c and Supplementary Fig. 3; CCDC: 2279440).

When *alliS*, *alliM*, and *alliB* were introduced into *A. oryzae*, the transformant with the three genes produced compound **8** (Fig. 3b, trace ix). In contrast to the products obtained from the other BGCs, **8** was identified as a pentacyclic onoceroid and named alliaonoceroid A (Fig. 3c). Although **8** has never been isolated from natural sources, it was predicted to be a biosynthetic precursor of cupacinoxepin[42], which had been isolated from the Ecuadorian plant *Cupania cinerea*[43]. We investigated steps involved in the biosynthetic processes catalyzed by the P450 AlliP and the acetyltransferase AlliA. We created four gene expression systems with either *alliP* or *alliA* to determine the enzyme required first in the biosynthetic process. Both the transformants yielded a specific product. The introduction of *alliP* and *alliA* led to the formation of **9** and **10**, respectively (Fig. 3b, traces x and xi), which were characterized to be the C-21 keto and C-6α hydroxy analogue of **8** and the *O*-acetylated form of **8**, respectively, through NMR and X-ray crystallographic analyses (Fig. 3c and Supplementary Fig. 3; CCDC: 2279441 and 2279442). Finally, the transformant harboring all five *alli* genes produced **11** (Fig. 3b, trace xii), which was determined to possess two acetoxy groups at the C-16β and C-21α positions (Fig. 3c and Supplementary Fig. 3; CCDC: 2279443). Compounds **9**–**11** were designated as alliaonoceroids B–D, respectively.

We then investigated whether the fungal strains that originally harbor the onoceroid BGCs yield the onoceroid species obtained in this study. We found that, based on LC–APCI–MS/MS analysis, *A. homomorphus* CBS 101889 and *A. alliaceus* CBS 536.65 produced **4** and **11**, respectively (Supplementary Fig. 4), which are the predicted end pathway products of the *homo* and *alli* clusters, respectively. Accordingly, the expression of the biosynthetic genes was also confirmed by RT-PCR (Supplementary Fig. 5). Meanwhile, the metabolite analysis of *A. fumigatus* CBS 144.89 suggested that **7**, the product from the *fumi* cluster, is formed by the fungus when it was cultivated on potato dextrose agar (PDA) plate (Supplementary Fig. 4); however, the MS/MS spectrum of the concerned compound could not be obtained due to its low productivity, and therefore, we were unable to confirm the production of **7** in *A. fumigatus*.

## Discovery of 4-hydroxybenzoate-derived meroterpenoids
To further demonstrate the utility of our genome mining strategy, we then aimed to discover another type of fungal metabolites by characterizing an atypical BGC that is not detected by antiSMASH. To this end, we focused on a BGC derived from *Neoarthrinium moseri* CBS 164.80, which was designated as the *mos* cluster (Figs. 2b and 4a and Supplementary Table 3). The *mos* cluster encoded the two FAD-dependent monooxygenases/oxidoreductases MosA and MosD, the Pyr4-family terpene cyclase MosB, the UbiA-like prenyltransferase MosC, the *N*-acetyltransferase MosE, and the hypothetical protein MosF. To obtain the metabolite synthesized by the *mos* cluster, the six *mos* genes were heterologously expressed in *A. oryzae* NSARU1[44]. The resultant *A. oryzae* transformant yielded metabolites **12** and **13**, which were absent in the host fungal strain (Fig. 4b, traces i and ii), both of which possess the molecular formula $C_{21}H_{32}O_3$. However, the productivity of the two metabolites was considerably low, probably due to the insufficient supply of their precursor molecule. Interestingly, we noted that the FMO MosD is highly homologous to the

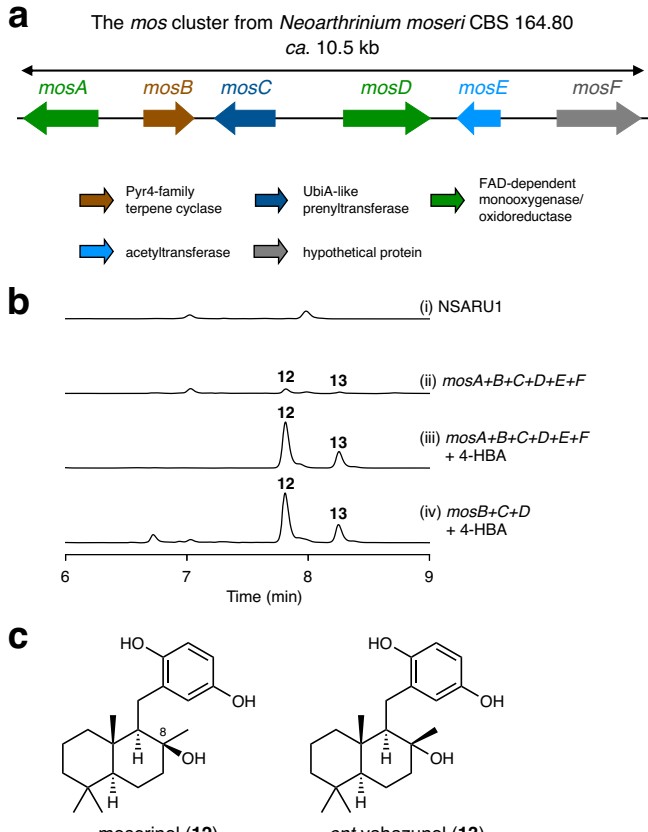

**Fig. 4 | Characterization of the BGC for 4-hydroxybenzoate-derived meroterpenoids. a** Schematic representations of the *mos* cluster. **b** High-performance liquid chromatography profiles of the metabolites obtained from *Aspergillus oryzae* transformants. The chromatograms were monitored at 300 nm. **c** Structures of meroterpenoids obtained in this study.

4-hydroxybenzoate decarboxylase BisD (60% amino acid sequence identity), which is involved in the biosynthesis of biscognienyneB[45]. Thus, we reasoned that 4-hydroxybenzoic acid (4-HBA) served as the precursor for **12** and **13**. To investigate this hypothesis, the *A. oryzae* transformant was cultivated in the presence of 4-HBA, which resulted in the production of **12** and **13** in a much higher yield (Fig. 4b, trace iii). Subsequently, both compounds were isolated from large-scale cultivation. The minor product **13** was identified as *ent*-yahazunol based on its NMR spectra and specific rotation (Fig. 4c)[46,47], which was originally isolated from a sponge of the genus *Dysidea*. The NMR spectra of **12** revealed that **12** possesses the same planar structure as **13** and is the C-8 epimer of **13**, and **12** was named moserinol (Fig. 4c). As discussed later, the structures of **12** and **13** suggested that MosB, MosC, and MosD are sufficient to synthesize these compounds. Thus, we created another *A. oryzae* transformant harboring *mosB*, *mosC*, and *mosD*, which expectedly afforded **12** and **13** (Fig. 4b, trace iv). The metabolite analysis of *N. moseri* CBS 164.80 revealed the presence of **12** upon feeding 4-HBA (Supplementary Fig. 4), and the RT-PCR analysis confirmed the expression of all *mos* genes except for *mosA* (Supplementary Fig. 5).

We then evaluated the antibacterial activity of compounds **12** and **13** by a disk diffusion assay using a 5 mm paper disk loaded with 25 μg of each compound. As a result, only **12** exhibited weak activities against *Bacillus cereus*, *Staphylococcus epidermidis*, *Staphylococcus faecalis*, and *Staphylococcus aureus* (11, 11, 9, and 11 mm of inhibition zones, respectively; lengths are the averages from three independent experiments). Meanwhile, ampicillin, which was used as a positive control, formed 23, 29, 34, and 47 mm of inhibition zones against the

four bacterial species, respectively, under the same experimental condition.

## Discussion

In this study, we developed a fungal genome mining tool, which was based on a manually curated fungal BGC database and custom-made HMM profiles. We demonstrated that our genome mining tool could effectively identify BGCs for previously undiscovered natural products. The onoceroid BGCs characterized in this study could also be identified using antiSMASH; however, the BGCs extracted by anti-SMASH contained substantially more genes outside the boundary of a BGC than those extracted in this study (Supplementary Fig. 6). In addition, antiSMASH could not detect the presence of *pyr4*-like genes in these BGCs. Alternatively, these BGCs could be discovered by standard BLAST search using Pyr4 as a query and subsequent manual investigation of the flanking regions of each identified *pyr4*-like gene, which, however, requires tedious procedures and might take weeks to months for completion. Therefore, the onoceroid BGCs would not have been effectively discovered using antiSMASH or conventional genome mining methodologies. On the other hand, our genome mining tool has allowed the automated extraction and visualization of all possible BGCs encoding a Pyr4 homologue from approximately 2000 fungal genomes within a single day, which also facilitated the selection of BGCs with unusual features. Although we focused only on BGCs with a *pyr4*-like terpene cyclase gene in this study, our strategy can be readily applied to extract BGCs that encode a homologue of user-selected proteins. For example, a similar global genome mining analysis can be conducted to extract BGCs encoding a homologue of PydY, which is a domainless enzyme and serves as the pericyclase in the biosynthesis of pyrrocidines (Supplementary Fig. 7)[48]. PydY and its homologues, namely ScpY[48], PN3-20[49], G73[50], and MGG_15096[51], have been found in the biosynthetic pathways of a few fungal polyketide–nonribosomal peptide hybrid molecules; however, their prevalence in fungal natural product pathways has not been investigated. The global genome analysis resulted in the detection of more than 300 BGCs with a *pydY* homologue (Supplementary Fig. 7 and Supplementary Data 5), some of which displayed interesting features, including those with an SHC/OSC gene and others with both UbiA-like prenyltransferase and polyprenyl pyrophosphate synthase genes. Intriguingly, one of the identified BGCs encodes a Pyr4 homologue and is of chytrid origin, which was mentioned earlier (Fig. 2b). The PydY homologues encoded by these BGCs might possess a catalytic function other than as a pericyclase. Collectively, our genome mining strategy has general applicability and will facilitate and accelerate the discovery of unexploited natural products, particularly those synthesized with the involvement of a domainless enzyme.

In our global fungal genome mining study, we discovered three types of onoceroids, which are triterpenoids synthesized from squalene or oxidosqualene through cyclization at both ends of the prenyl chain. Onoceroids have been isolated from bacteria[52], ferns[53], higher plants[54,55], and animals[56]. However, to the best of our knowledge, onoceroids have never been isolated from fungi. Thus, the present study provides landmark examples of fungal onoceroids and their biosynthetic pathways. In terms of the biosynthesis of onoceroids, BmeTC from the bacterium *Bacillus megaterium* was the initial onoceroid synthase characterized in 2013, and this enzyme solely transforms squalene into onoceroids[52]. In the fern *Lycopodium clavatum*, a pair of homologous enzymes (LCC and LCD or LCE) convert oxidosqualene into onoceroid species[57,58]. These known enzymes for onoceroid biosynthesis are all homologous to SHCs and OSCs. In contrast, the biosynthesis of fungal onoceroids requires two families of terpene cyclases (i.e., SHC/OSC-like enzyme and Pyr4-family terpene cyclase), introducing an unprecedented biosynthetic mechanism of onoceroids. This study demonstrated that Pyr4-family terpene cyclases are also involved in the biosynthesis of pure (not mero-) terpenoids. The

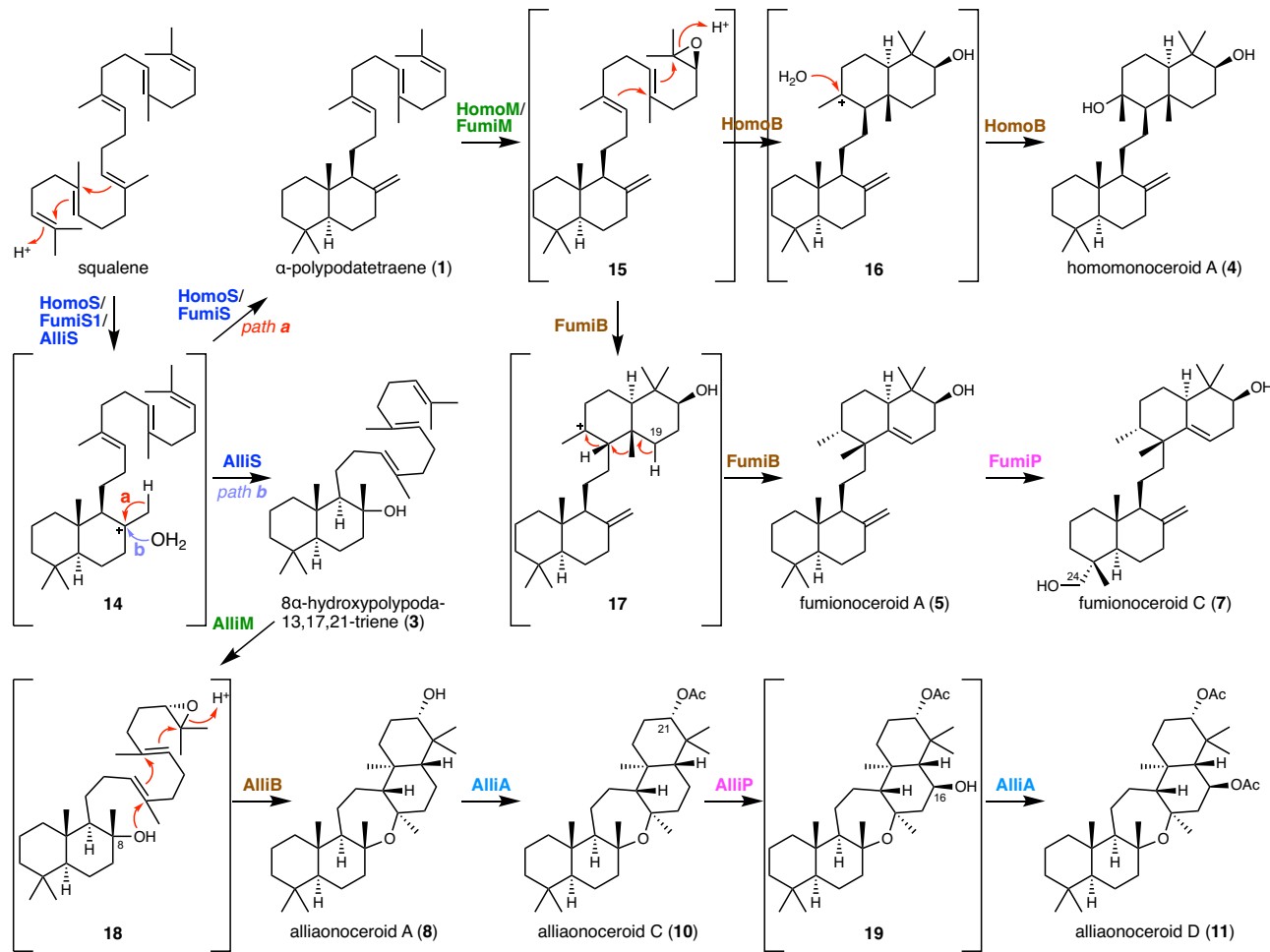

**Fig. 5 | Proposed biosynthetic pathways of the onoceroids obtained in this study.** Predicted biosynthetic/reaction intermediates are shown in brackets.

majority of fungal triterpenoids are synthesized through the cyclization of oxidosqualene[59–61], and hexaprenyl pyrophosphate is known to serve as the precursor of a few fungal triterpenoids[62]. However, the fungal onoceroid pathways identified in this study represent rare examples in which squalene is directly cyclized to produce fungal triterpenoids.

The biosynthetic pathways of fungal onoceroids discovered in this study can be proposed as follows (Fig. 5). The biosynthesis of homomonoceroid A (**4**) begins with squalene being cyclized into α-polypodatetraene (**1**) by the squalene cyclase HomoS through the carbocationic intermediate **14**. Subsequently, the FMO HomoM epoxidizes the terminal olefin of **1** to yield epoxide **15**, which was not isolated in this study. The Pyr4-family terpene cyclase HomoB then protonates epoxide **15** to initiate the second round of cyclization, and the resulting tetracyclic carbocationic species **16** is neutralized by a water attack, yielding **4**. In addition, the bicyclic intermediate **15** is involved in the biosynthesis of fumionoceroid C (**7**), where FumiB accepts **15** to produce a differently cyclized product fumionoceroid A (**5**). FumiB first cyclizes **15** into the carbocationic intermediate **17**. Instead of being quenched by water, the reaction concludes with a 1,2-hydride shift, 1,2-methyl shift, and deprotonation from C-19. Subsequently, the P450 FumiP hydroxylates **5** at C-24 to complete the biosynthesis. Oxidosqualene can also be accepted by Fumi enzymes, except for FumiP, to produce the C-3 hydroxy analogue of **5**, fumionoceroid B (**6**), through α-polypodatetraen-3β-ol (**2**) (Supplementary Fig. 8). Meanwhile, the biosynthetic pathway leading to alliaonoceroid D (**11**) branches from the other two pathways in the first step. AlliS transforms squalene into 8α-hydroxypolypoda-13,17,21-triene (**3**)

instead of **1**. After the epoxidation of **3** by AlliM to yield **18**, which was not obtained in our work, AlliB cyclizes **18** into alliaonoceroid A (**8**). The cyclization mode followed by AlliB is similar to that employed by HomoB, but the AlliB-catalyzed reaction uses the C-8 hydroxy group instead of a water molecule to produce the pentacyclic onoceroid **8**. Compound **8** then undergoes acetylation at the C-21 hydroxy group; this reaction is catalyzed by the acetyltransferase AlliA, yielding alliaonoceroid C (**10**). The P450 AlliP then installs a hydroxy group at C-16 to produce **19**, which is again acetylated by AlliA to yield the end product **11**. In the absence of AlliA, AlliP might cause the hydroxylation of **3** at C-6 to yield a shunt pathway product **19**, which is then oxidized by an endogenous enzyme of *A. oryzae* to give alliaonoceroid B (**9**) (Supplementary Fig. 8). The three characterized fungal onoceroid pathways suggest that fungi synthesize diverse onoceroids due to various factors affecting structural diversification, such as the (i) cyclization mode of (oxido)squalene, (ii) the reaction mechanisms of Pyr4-family terpene cyclases, and (iii) tailoring reactions. The characterization of other onoceroid BGCs not examined in this study or the construction of artificial pathways with onoceroid biosynthetic genes from different pathways can further expand the molecular diversity of fungal onoceroids.

Finally, we obtained two 4-HBA-derived meroterpenoids by the heterologous expression of the *mos* cluster, which was not detectable by antiSMASH. The major product from the *mos* cluster, moserinol (**12**), was determined to be a previously unreported molecule, further demonstrating the utility of our genome mining tool in accessing unexploited compounds. Unfortunately, **12** lacks a structural novelty, as its diastereomers, yahazunol and *ent*-yahazunol (**13**), have already

been isolated from nature[46,63]. However, these two compounds have been obtained from a brown alga and a sponge, respectively, and, to the best of our knowledge, these sesquiterpene hydroquinones, and their analogues, have never been reported from fungi. In addition, although 4-HBA is a precursor for diverse meroterpenoids both from primary and secondary metabolism, such as ubiquinones[64], xiamenmycin[65], and biscognienyneB[45], our study provides an intriguing example in which a Pyr4-family terpene cyclase is involved in the biosynthesis of 4-HBA-derived meroterpenoids. Although the complete biosynthetic pathway of **12** and the functions of each Mos protein await to be elucidated in future studies, the biosynthetic route to **12** could be proposed as follows (Supplementary Fig. 9). Initially, the UbiA-like prenyltransferase MosC farnesylates 4-HBA, which is followed by the oxidative decarboxylation by the FMO MosD. Subsequently, the terpene cyclase MosB cyclizes the farnesyl moiety, in which the cyclization is completed by the attack of a water molecule from both sides, resulting in the formation of a pair of epimers, **12** and **13**. The proposed pathway is consistent with the fact that *mosB*, *mosC*, and *mosD* are sufficient to yield the two meroterpenoids (Fig. 4b, trace iv). It should also be mentioned that our recent study based on the genome mining analysis reported herein unraveled another unprecedented biosynthetic mechanism for fungal meroterpenoids, in which Pyr4 homologues cyclize the prenyl moiety installed by a dimethylallyltryptophan synthase (DMATS)-type prenyltransferase[66]. Overall, our global genome mining has so far identified three previously undescribed forms of biosynthetic mechanisms with Pyr4-family terpene cyclases.

In conclusion, we successfully isolated several previously unreported natural products using our genome mining tool. We believe that this tool can be widely applied for the discovery of natural products with unprecedented scaffolds. Currently, our genome mining tool mainly targets known–unknown BGCs[67], which produce unknown natural products synthesized by the known classes of core enzymes. Recent studies have highlighted the genome mining-driven discovery of unknown–unknown natural products[68]. We are now enhancing the genome mining platform by incorporating additional functions to readily extract BGCs encoding self-resistance enzymes[26] or BGCs that lack a known core protein. This would facilitate the discovery of unexploited bioactive or unknown–unknown natural products from fungi.

## Methods
### General experimental procedures
Organic solvents were purchased from Anaqua (Hong Kong) Co. Ltd., and other chemicals were purchased from Wako Chemicals Ltd., Thermo Fisher Scientific, Sigma–Aldrich, or J&K Scientific Ltd., unless noted otherwise. Oligonucleotide primers (Supplementary Data 9) were purchased from Beijing Genomics Institute. PCR was performed using a T100 Thermal Cycler (Bio-Rad Laboratories, Inc.) with Phanta Max Super-Fidelity DNA Polymerase (Vazyme Biotech Co., Ltd). GC–MS analyses were performed with an Agilent 7890 A/5975 C system using an HP-5ms capillary column (0.25 mm i.d. × 30 m, 0.25 μm film thickness). Preparative HPLC was performed on a Waters 1525 Binary HPLC pump with a 2998 photodiode array detector (Waters Corporation), using a COSMOSIL 5C18-AR-II column (10 i.d. x 250 mm, Nacalai Tesque, Inc). Flash chromatography was performed using an Isolera Spektra One flash purification system (Biotage). NMR spectra were obtained at 600 MHz ($^1$H)/150 MHz ($^{13}$C) with a Bruker Ascend Avance III HD spectrometer at 25 °C, and chemical shifts were recorded with reference to solvent signals ($^1$H NMR: CDCl$_3$ 7.26 ppm, C$_6$D$_6$ 7.16 ppm, acetone-$d_6$ 2.05 ppm; $^{13}$C NMR: CDCl$_3$ 77.0 ppm, C$_6$D$_6$ 128.06 ppm, acetone-$d_6$ 29.8 ppm). HR-APCI-MS and HR-ESI-MS spectra were obtained with a SCIEX X500R Q-TOF mass spectrometer. Samples for LC-MS analysis were injected into a SCIEX ExionLC AD System with a SCIEX X500R Q-TOF mass spectrometer, using an

Accucore C18 column (4.6 i.d. x 100 mm; Thermo Scientific) for triterpenoids and a Luna Omega 1.6 μm C18 100 Å column (2.1 i.d. x 100 mm; Phenomenex) for sesquiterpene hydroquinones. Optical rotations were measured with a P-2000 Digital Polarimeter (JASCO Corporation). X-ray diffraction data were collected on a Bruker D8 Venture Photon II diffractometer. UV spectra were measured using a LAMBDA 1050 + UV/Vis/NIR Spectrophotometer (PerkinElmer). IR data were recorded on a Spectrum 100 FT-IR Spectrometer (PerkinElmer).

### Microbial strains
*Aspergillus alliaceus* CBS 536.65, *Aspergillus fumigatus* CBS 144.89, *Aspergillus homomorphus* CBS 101889, and *Neoarthrinium moseri* CBS 164.80 were purchased from the Westerdijk Fungal Biodiversity Institute. *Aspergillus oryzae* NSAR1 (*niaD⁻, sC⁻, ΔargB, adeA⁻*)[34] and NSARU1 (*niaD⁻, sC⁻, ΔargB, adeA⁻, pyrG⁻*)[44] were utilized as the fungal heterologous expression host. Standard DNA engineering was performed with *Escherichia coli* DH5α (Takara Bio Inc).

### Collection and curation of fungal biosynthetic gene clusters
When genomic sequence data corresponding to a fungal BGC were publicly available, we obtained these data from the NCBI or the Joint Genome Institute (JGI) databases. For sequence data lacking gene annotations, gene prediction was performed using AUGUSTUS[69] (version 3.5.0). All of the gene products were compared with their homologous proteins available in the NCBI database, and their sequences were manually revised based on this comparison when necessary. All of the BGC sequences were stored in a GenBank format, with each biosynthetic protein represented as a coding DNA sequence (CDS) feature and assigned a unique identification number that starts with FBGC (e.g., FBGC00001). Each CDS feature was labeled with its gene name and given a product qualifier consisting of the protein classification and protein name (e.g., terpene cyclase Pyr4).

In cases where the complete BGC sequence was unavailable but individual gene or protein sequence data were available, all of the genes/proteins for a BGC were collected in a directory with a unique identification number starting with FPROT (e.g., FPROT00001).

Information on all of the collected BGCs is summarized in Supplementary Data 1, which comprises their representative metabolite, source organism, and primary reference. The FBGC GenBank files and FPROT directories were deposited at Zenodo under the DOI 10.5281/zenodo.8126803. The database was named FunBGCs (Collection of Manually Curated Fungal Biosynthetic Gene Clusters) and is available online at http://staffweb1.cityu.edu.hk/ymatsuda/funbgcs/funbgcs.html.

### Development of a fungal genome mining tool
**Creation of a custom Pfam database for fungal biosynthetic protein detection.** All of the protein sequences (5,070 proteins) were initially extracted from the collected fungal BGCs and analyzed using the hmmscan tool in HMMER software[17] (version 3.3.2). The analysis was conducted against the Pfam-A HMM library (downloaded from https://ftp.ebi.ac.uk/pub/databases/Pfam/current_release/Pfam-A.hmm.gz) using an E-value and conditional E-value cutoff of 1e-5. We detected 520 Pfam domains, which were used for subsequent bioinformatic analyses.

No Pfam domain was detected in 572 proteins, and these protein sequences were employed to create a BLAST database using the makeblastdb function in NCBI BLAST + [70] (version 2.11.0). The 572 proteins were classified into groups based on the findings of BLASTp analysis (E-value cutoff: 0.01) using the aforementioned created database. If a protein aligned with another protein in the BLASTp analysis, then these two proteins were placed in the same group. If a protein group had more than one protein, then sequence alignment was performed using Clustal Omega[71] (version 1.2.3). These sequence alignments were then converted into an HMM file using the hmmbuild tool in HMMER. Each manually created HMM file was named after a

representative member of the respective protein group (e.g., Pyr4.hmm). This process yielded 59 additional HMM profiles.

For the enhanced detection of PKS/NRPS domains, some HMM profiles were adopted from the SMART[23] and TIGRFAMs[24] databases. Furthermore, several HMM files for PKS/NRPS domains were manually created using the PKSs and NRPSs available in the FunBGCs database. Several Pfam HMM files were replaced with these additional HMM files. All of the HMM profiles used in this study are summarized in Supplementary Data 2 with their sources and accession numbers. The manually created HMM files were deposited at Zenodo under the DOI 10.5281/zenodo.8126803, and their descriptions are provided in Supplementary Data 3.

**Extraction of BGCs of interest from given fungal genome sequence data.** To extract BGCs encoding the homologues of proteins of interest from a given fungal genome (which needed to be provided as a GenBank file containing CDS features with a translation qualifier), the following steps were implemented. First, all protein sequences from the genome were extracted and then analyzed using hmmscan against the HMM profile created from the proteins of interest; an E-value and conditional E-value cutoff of 1e-5 were applied.

Subsequently, products from 20 genes (if available) in the flanking regions of each gene identified in the previous step were extracted. The protein sequences were analyzed using hmmscan against the HMM library created for the detection of core enzymes (refer to Supplementary Table 1 for detection criteria). We then conducted a search for possible precursor peptides involved in the biosynthesis of ribosomally synthesized and posttranslationally modified peptides (RiPPs)[72]. For this purpose, a protein was considered a candidate for a RiPP precursor peptide (RiPP PP) if its sequence contained more than two partial sequences that were ≥90% identical, with each containing ≥10 amino acids, with ≥3 different amino acids, and either KR, KK, RK, or RR. The extracted protein sequences were analyzed using hmmscan against the HMM library containing all HMM profiles for fungal biosynthetic proteins; an E-value and conditional E-value cutoff of 1e-5 were applied. In addition, a BLASTp search was performed using DIAMOND[25] (version 2.1.7) to compare the extracted proteins against the protein database containing all proteins from the FunBGCs database. Furthermore, we looked for the presence of a close homologue of each extracted protein with ≥50% identity in the given fungal genome; some BGCs are known to encode a self-resistant enzyme that is a close homologue of a housekeeping protein but is a resistant version that is not inhibited by the metabolite synthesized by the BGC[26]. Proteins identified in at least one of the aforementioned analyses were classified as biosynthetic proteins. Small proteins with <200 amino acid residues were removed from the subsequent analysis unless they were classified as biosynthetic proteins.

Next, we predicted whether two adjacently located genes were clustered, indicating that they were responsible for the biosynthesis of the same metabolite. We performed a BLASTp analysis of the two proteins with DIAMOND and used the protein database containing all of the proteins from the FunBGCs database. BLAST hits with ≥25% identity and ≥50% sequence coverage were used in the step. If the origin (parental BGC) of a protein aligned with one of the query proteins was identical to that of at least one of the proteins aligned with the other query and the distance between the two genes encoding the two proteins was less than a set length (default: 15,000 bp), then the two proteins were considered clustered. However, as an exception, if the two proteins both aligned with a protein from the same origin with ≥50% identity, then they were considered clustered regardless of the distance between the genes encoding the two proteins. If two adjacently located genes were not considered clustered, then the same analysis was performed on the next gene until no correlation was

observed in three consecutive genes. If two genes were considered clustered, all of the genes between the two genes were also considered clustered.

To extract BGCs from each scaffold, genes were individually examined from one end to identify those encoding a biosynthetic protein (as defined above). When a gene encoding a biosynthetic protein was identified, it was considered a starting point of a possible BGC. Subsequent genes were included in the BGC as long as they met the following criteria: a gene of interest (i) was considered clustered with the previous gene or (ii) encoded a biosynthetic protein and was closely located with the previous gene (default: within 2500 bp). If such a candidate BGC encoded a core enzyme or both a RiPP precursor peptide candidate and a UstY-like protein, then the BGC was extracted as a GenBank file.

**Extraction of all BGCs from given fungal genome sequence data.** The aforementioned methods can be used to extract all possible BGCs from a given fungal genome with slight modifications, although these modified methods were not used in the current study. After the extraction of all of the proteins, we searched for core enzymes and RiPP precursor peptides instead of identifying the homologues of target proteins. Other BGC extraction procedures were the same as described above.

**Calculation of the similarity score.** To examine whether a given BGC was similar to a known BGC, the similarity score was calculated using the following Eq. (1):

$$similarity\ score = [average\ identity\%] \times \frac{[number\ of\ homologous\ genes\ in\ a\ given\ BGC]}{[number\ of\ total\ genes\ in\ a\ known\ BGC]}$$
(1)

where the average identity% is the average percentage value of the identities of each pair of corresponding proteins in the two BGCs. A pair of proteins with ≥45% identity and ≥50% sequence coverage was considered homologous in this calculation.

The genome mining tool developed herein was designated as FunBGCeX (Fungal Biosynthetic Gene Cluster eXtractor). FunBGCeX has been deposited at Zenodo under the DOI 10.5281/zenodo.8126797 and is also available at https://github.com/ydmatsd/funbgcex.

## Extraction of biosynthetic gene clusters encoding a Pyr4 or PydY homologue

To extract BGCs encoding a Pyr4 homologue from available fungal genomes, we downloaded 1990 annotated fungal reference genomes from the NCBI database. BGCs were extracted using the HMM profile for Pyr4 homologues (Pyr4.hmm) with default parameters. This process yielded 1050 BGCs (including those with a single gene). Information on all the extracted BGCs is provided in Supplementary Data 5. The extraction of BGCs with a *pydY*-like gene was performed in a similar manner to the HMM profile for PydY homologues (PydY.hmm).

## Phylogenetic analysis of Pyr4 homologues

The sequences of known Pyr4-family terpene cyclases and their homologues identified in this study were first aligned using MUSCLE[73] (version 5.1), and the conserved sequences were extracted with Gblocks[74] (version 0.91b). A maximum likelihood (ML) phylogenetic tree was generated with FastTree[75] (version 2.1.11), and the resultant phylogeny was used as a starting tree to infer an ML tree using RaxML[76] (version 8.0.0) under the LG + G + F model, as identified by ProtTest[77] (version 3.4.2). The Pyr4 homologues of bacterial origin were used as outgroups. The phylogenetic tree was visualized with Geneious Prime 2023.2.1 (https://www.geneious.com).

## Analysis of fungal genomes using antiSMASH

The 1990 fungal genomes mentioned above were analyzed by the standalone version of antiSMASH (version 6.1.1) using the default parameters. Subsequently, all protein sequences from all the detected BGCs were extracted and analyzed using hmmscan against the HMM profile of Pyr4 to extract BGCs with a *pyr4* homologue; an E-value and conditional E-value cutoff of 1e-5 were applied.

## Construction of fungal transformation plasmids

Initially, each gene in the *homo*, *fumi*, *alli*, and *mos* clusters was amplified from the genomic DNA of *A. homomorphus* CBS 101889, *A. fumigatus* CBS 144.89, and *A. alliaceus* CBS 536.65, and *N. moseri* CBS 164.80 with the primers shown in Supplementary Data 9 and Supplementary Table 4, and then ligated to the pTAex3-HR vector[78] or pPyrG-HR vector[44], using a ClonExpress Ultra One Step Cloning Kit (Vazyme Biotech Co., Ltd). To construct fungal transformation plasmids with multiple genes, DNA fragments with the *amyB* promoter (PamyB) and the *amyB* terminator (TamyB) were amplified from the pTAex3-HR-based plasmids, and further introduced into the already constructed single gene-containing vector or another vector, pAdeA-HR[79]. Details of the plasmid construction are provided in Supplementary Table 4.

## Fungal transformation

Transformation of *A. oryzae* NSAR1 was performed by the protoplast–polyethylene glycol method[13] coupled with CRISPR–Cas9-guided homologous recombination[44,78,79]. Initially, *A. oryzae* NSAR1 or NSARU1 was cultivated in 10 mL of DPY medium [2% dextrin, 1% hipolypepton (Nihon Pharmaceutical Co., Ltd.), 0.5% yeast extract, 0.5% $KH_2PO_4$, 0.05% $MgSO_4 \cdot 7H_2O$] supplemented with 0.01% adenine, 0.2% uracil, and 0.5% uridine (uracil and uridine were only added for the NSARU1 strain) for two days at 30 °C and at 160 rpm. The resulting preculture was then transferred to 100 mL of DPY medium with the necessary supplements and incubated at 30 °C and 160 rpm for 24 h. Mycelia were then collected and incubated in TF solution 1 [1% Yatalase (Takara Bio Inc.) and 0.6 M $(NH_4)_2SO_4$ in 50 mM maleic acid (pH 5.5)] for 2 h at 30 °C and at 160 rpm. The resultant protoplast-containing mixture was filtrated, and the filtrate was centrifuged at 1,500 rpm for 10 min at room temperature. The protoplast pellet was washed with TF solution 2 (10 mM Tris-HCl, pH 7.5, 1.2 M sorbitol, 50 mM $CaCl_2$, 35 mM NaCl) and resuspended in TF solution 2 to a concentration of $1–5 \times 10^7$ cells/mL. Meanwhile, Cas9 ribonucleoprotein (RNP) complexes were assembled as follows: (i) dissolve lyophilized Alt-R CRISPR-Cas9 RNA [crRNA; Integrated DNA Technologies (IDT)] [5'-AGTGTTGC AATCCAAGGATA-3' (for the HS801 locus[80]), 5'-CAAGGACCACG TCCTTCAAC-3' (for the *sC* locus[81]), 5'- TGTCGGAAGTTTAGTACCAA-3' (for the HS401 locus[80])] and trans-activating crRNA (tracrRNA; IDT) in the nuclease-free duplex buffer to a concentration of 100 μM; (ii) mix each crRNA separately with equal volumes of tracrRNA and nuclease-free duplex buffer, boil the resulting mixtures at 95 °C for 5 min, and then cool at room temperature for 15 min to generate guide RNA (gRNA); (iii) dilute the Alt-R S.p. Cas9 Nuclease V3 (IDT) solution with 1 × phosphate-buffered saline (PBS) to a final concentration of 1 μg/μL; and (iv) combine 1.5 μL of each gRNA solution with 0.75 μL of Cas9 (1 μg/μL) and 11 μL of 1 × PBS (final volume: 13.25 μL), and incubate the resulting mixture at room temperature for 5 min to generate dual RNP complexes. Subsequently, 5 μg of each plasmid, 13.25 μL of the corresponding dual RNP complexes, and 400 μL of protoplast suspension were mixed and incubated at room temperature for 30 min. A total of 2.7 mL of TF solution 3 (10 mM Tris-HCl, pH 7.5, 60% PEG4000, 50 mM $CaCl_2$) was then added in three portions (500, 500, and 1700 μL, respectively) to the resulting mixture. After 20 min incubation at room temperature, the mixture was diluted with 10 mL of TF solution 2 and then centrifuged at 1,500 rpm for 10 min at room temperature. The protoplast-cell pellet was resuspended in 500 μL of TF solution 2 and then mixed with 5 mL of M-sorbitol [0.2% $NH_4Cl$, 0.1% $(NH_4)_2SO_4$,

0.05% KCl, 0.05% NaCl, 0.1% $KH_2PO_4$, 0.05% $MgSO_4 \cdot 7H_2O$, 0.002% $FeSO_4 \cdot 7H_2O$, 2% glucose, 1.2 M sorbitol, and supplemented with 0.1% arginine, 0.15% methionine, 0.01% adenine, when necessary, pH 5.5] top agar medium (1% agar), which was poured on the top of the pre-made M-sorbitol bottom agar plate (2% agar). The plate was further incubated for 3–7 days at 30 °C until transformant colonies appeared. Finally, the resultant colonies were transferred to M agar plates with the necessary supplements (and without sorbitol) and incubated for 2–3 days. The transformants created in this study and the plasmids used for the transformation are provided in Supplementary Table 5.

The successful construction of a transformant was confirmed by PCR-amplifying each gene in the transformant (Supplementary Fig. 10). To confirm the expression of the biosynthetic genes, the *A. oryzae* transformants with *homoS* + M + B, *fumiS1* + B + M + P, *alliS* + M + B + P + A, or *mosA* + B + C + D + E + F were cultivated in DPY liquid medium [2% dextrin, 1% hipolypepton (Nihon Pharmaceutical Co., Ltd.), 0.5% yeast extract, 0.5% $KH_2PO_4$, and 0.05% $MgSO_4 \cdot 7H_2O$], and their total RNA was extracted by using a GeneJET Plant RNA Purification Kit (Thermo Scientific). Subsequently, complementary DNA (cDNA) synthesis was performed by using a HiScript III 1st Strand cDNA Synthesis Kit ( + gDNA wiper) (Vazyme Biotech), and the expression of each gene was confirmed by PCR using the synthesized cDNA as a template (Supplementary Fig. 11).

## Analysis of metabolites derived from *A. oryzae* transformants

To analyze the metabolites produced by each *A. oryzae* transformant, the transformants were cultivated on a DPY agar plate (1.5% agar) for seven days at 30 °C. When necessary, 0.01% 4-HBA was supplemented to the DPY plate. After cultivation, a small piece of fungal mycelia and agar was cut from the plate, soaked in ethyl acetate, and extracted using an ultrasonic bath. The ethyl acetate layer was transferred to another tube, and the solvent was removed using nitrogen gas flow. The residue was dissolved in methanol and then subjected to a GC–MS or an HPLC analysis. For the GC–MS analysis, the temperature of the ionization chamber was 260 °C, with electron impact ionization at 70 eV. Helium was used as a carrier gas, and its average velocity was 29.374 cm/sec. The initial temperature of the program was 150 °C, and it increased to 250 °C at a rate of 40 °C/min, then increased at a rate of 5 °C/min to 320 °C and held at 320 °C for 5 min. The HPLC analysis was performed with a solvent system of 20 mM formic acid (solvent A) and acetonitrile containing 20 mM formic acid (solvent B), at a flow rate of 0.4 mL/min and a column temperature of 40 °C. Separation was performed using a linear gradient from 10:90 (solvent B/solvent A) to 100:0 for 10 min, 100:0 for the following 3 min, and a linear gradient from 100:0 to 10:90 within the following 2.0 min, and then 10:90 for 2.5 min of equilibrium.

## Isolation of each metabolite from *A. oryzae* transformants

To isolate each metabolite, *A. oryzae* transformants were cultivated either on DPY agar plates (the volume of medium in one plate was *ca.* 20 mL) for seven days at 30 °C or in DPY liquid medium at 30 °C/ 160 rpm for five days.

When cultivated on the agar plates, the resultant fungal cultures, including agar medium, were crushed into small pieces, soaked in acetone, and extracted three times using an ultrasonic bath. After filtration, acetone was removed in vacuo. The residue was then partitioned between water and ethyl acetate, and the water layer was further extracted with hexane. Both ethyl acetate and hexane extracts were combined, fractionated by flash chromatography, and then purified by open silica-gel column chromatography or preparative HPLC.

When cultivated in the liquid medium, medium and mycelia were first separated by filtration. The mycelia were extracted with acetone with sonication for one hour, concentrated, and reextracted with ethyl acetate. The resultant crude extract was fractionated by flash chromatography, and further purified by preparative HPLC.

The purification methods for each compound are described in detail below.

**Purification conditions for α-polypodatetraene (1).** The extract of the *A. oryzae* strain with *homoS* cultivated on 50 DPY agar plates was subjected to flash chromatography and eluted stepwise using a hexane:ethyl acetate gradient (100:0 to 100:19). Fractions that contained **1** were then purified by open silica-gel column chromatography and eluted with hexane to yield 20.5 mg of **1**. Compound **1** (16.7 mg) was also isolated from the *A. oryzae* strain with *fumiS1* cultivated on 150 DPY agar plates, using a similar method.

**Purification conditions for α-polypodatetraen-3β-ol (2).** The extract of the *A. oryzae* strain with *fumiS1* cultivated on 150 DPY agar plates was subjected to flash chromatography and eluted stepwise using a hexane:ethyl acetate gradient (100:0 to 100:21). Fractions that contained **2** were then purified by preparative HPLC (100% acetonitrile, 3.0 mL/min) to yield 25.7 mg of **2**.

**Purification conditions for 8α-hydroxypolypoda-13,17,21-triene (3).** The extract of the *A. oryzae* strain with *alliS* cultivated on 50 DPY agar plates was subjected to flash chromatography and eluted stepwise using a hexane:ethyl acetate gradient (100:0 to 100:24). Fractions that contained **3** were then purified by preparative HPLC (100% acetonitrile, 3.0 mL/min) to yield 58.6 mg of **3**.

**Purification conditions for homomonoceroid A (4).** The extract of the *A. oryzae* strain with *homoS*, *homoB*, and *homoM* cultivated on 150 DPY agar plates was subjected to flash chromatography and eluted stepwise using a hexane:ethyl acetate gradient (100:0 to 100:24). Fractions that contained **4** were then purified by reverse-phase preparative HPLC (90% aqueous acetonitrile, 3.0 mL/min) to yield 80.5 mg of **4**.

**Purification conditions for fumionoceroids A (5) and B (6).** The extract of the *A. oryzae* strain with *fumiS1*, *fumiB*, and *fumiM* cultivated on 50 DPY agar plates was subjected to flash chromatography and eluted stepwise using a hexane:ethyl acetate gradient (100:0 to 100:31). Fractions that contained **5** were then purified by preparative HPLC (100% acetonitrile, 3.0 mL/min) to yield 21.2 mg of **5**. Fractions that contained **6** were then purified by preparative HPLC (100% acetonitrile, 3.0 mL/min) to yield 8.3 mg of **6**.

**Purification conditions for fumionoceroid C (7).** The extract of the *A. oryzae* strain with *fumiS1*, *fumiB*, *fumiM*, and *fumiP* cultivated on 50 DPY agar plates was subjected to flash chromatography and eluted stepwise using a hexane:ethyl acetate gradient (100:0 to 100:32). Fractions that contained **7** were then purified by preparative HPLC (95% aqueous acetonitrile, 3.0 mL/min) to yield 19.2 mg of **7**.

**Purification conditions for alliaonoceroids A (8) and B (9).** The extract of the *A. oryzae* strain with *alliS*, *alliB*, *alliM*, and *alliP* cultivated on 200 DPY agar plates was subjected to flash chromatography and eluted stepwise using a hexane:ethyl acetate gradient (100:0 to 100:17). Fractions that contained **8** were then purified by open silica-gel column chromatography and eluted with hexane:ethyl acetate (50:1) to yield 25.5 mg of **8**. Fractions that contained **9** were then purified by preparative HPLC (85% aqueous acetonitrile, 3.0 mL/min) to yield 18.8 mg of **9**.

**Purification conditions for alliaonoceroids C (10) and D (11).** The extract of the *A. oryzae* strain with *alliS*, *alliB*, *alliM*, *alliP*, and *alliA* cultivated on 200 DPY agar plates was subjected to flash chromatography and eluted stepwise using a hexane:ethyl acetate gradient (100:0 to 100:15). Fractions that contained **10** and **11** were then purified by open silica-gel column chromatography and eluted with hexane:ethyl acetate (120:1) to yield 21.5 mg of **10** and 15.3 mg of **11**.

**Purification conditions for moserinol (12) and *ent*-yahazunol (13).** The extract of the *A. oryzae* strain with *mosA*, *mosB*, *mosC*, *mosD*, *mosE*, and *mosF* cultivated in 3 L of DPY liquid medium was subjected to flash chromatography and eluted stepwise using a dichloromethane:ethyl acetate gradient (100:0 to 0:100). Fractions that contained **12** were then purified by reverse-phase preparative HPLC (60% aqueous acetonitrile, 3.0 mL/min) to yield 30.2 mg of **12**. Fractions that contained **13** were then purified by reverse-phase preparative HPLC (55% aqueous acetonitrile, 3.0 mL/min) to yield 12.8 mg of **13**.

### Synthesis of (*S*)-MTPA ester of compound 4

(*R*)-MTPA chloride (13.5 mg) was added to a solution of **4** (5.6 mg) in dry pyridine (1 mL) at room temperature. After 2 h, the reaction mixture was concentrated to dryness and purified by reverse-phase preparative HPLC (100% acetonitrile) to yield 7.1 mg of (*S*)-MTPA ester of **4**. $^1$H NMR (600 MHz, CDCl$_3$): δ [ppm] 4.83 (1H, brd, $J = 1.0$ Hz, H-26), 4.71 (1H, dd, $J = 11.7$, 4.4 Hz, H-21), 4.68 (1H, brs, H-26), 2.39 (1H, ddd, $J = 12.6$, 4.0, 2.3 Hz, H-7β), 1.97 (1H, td, $J = 12.7$, 4.8 Hz, H-7α), 1.86 (1H, dt, $J = 12.3$, 3.0 Hz, H-15β), 1.81 (1H, dq, $J = 12.7$, 4.1 Hz, H-20α), 1.74 (1H, m, H-1β), 1.72 (1H, m, H-6α), 1.68 (1H, m, H-19β), 1.65 (1H, m, H-16α), 1.63 (1H, m, H-20β), 1.60 (1H, m, H-12), 1.56 (1H, m, H-2β), 1.54 (1H, m, H-9), 1.51 (1H, m, H-2α), 1.49 (2H, m, H-11), 1.41 (1H, m, H-15α), 1.40 (1H, m, H-3β), 1.32 (1H, m, H-16β), 1.31 (1H, m, H-6β), 1.23 (1H, td, $J = 13.3$, 3.2 Hz, H-19α), 1.17 (1H, td, $J = 13.5$, 4.0 Hz, H-3α), 1.12 (3H, s, H-27), 1.07 (1H, dd, $J = 12.6$, 2.6 Hz, H-5), 1.03 (1H, t, $J = 3.5$ Hz, H-13), 1.02 (1H, dd, $J = 12.5$, 2.0 Hz, H-17), 0.97 (1H, td, $J = 12.9$, 3.8 Hz, H-1α), 0.92 (1H, m, H-12), 0.91 (3H, s, H-29), 0.87 (3H, s, H-24), 0.80 (3H, s, H-23), 0.78 (3H, s, H-30), 0.76 (3H, s, H-28), 0.65 (3H, s, H-25); for the NMR spectrum, see Supplementary Fig. 25.

### Synthesis of (*R*)-MTPA ester of compound 4

In the same manner as described for the (*S*)-MTPA ester of **4**, **4** (5.3 mg) was treated with (*S*)-MTPA chloride (13.5 mg) to yield 7.0 mg of (*R*)-MTPA ester of **4**. $^1$H NMR (600 MHz, CDCl$_3$): δ [ppm] 4.83 (1H, brd, $J = 0.9$ Hz, H-26), 4.74 (1H, dd, $J = 11.8$, 4.5 Hz, H-21), 4.68 (1H, brs, H-26), 2.39 (1H, ddd, $J = 12.7$, 4.0, 2.3 Hz, H-7β), 1.97 (1H, td, $J = 12.7$, 4.8 Hz, H-7α), 1.88 (1H, m, H-20α), 1.86 (1H, dt, $J = 12.2$, 3.0 Hz, H-15β), 1.77 (1H, dq, $J = 12.4$, 2.7 Hz, H-20β), 1.73 (1H, m, H-1β), 1.72 (1H, m, H-6α), 1.71 (1H, m, H-19β), 1.64 (1H, dq, $J = 13.3$, 2.2 Hz, H-16α), 1.60 (1H, m, H-12), 1.56 (1H, m, H-2β), 1.54 (1H, m, H-9), 1.49 (1H, m, H-2α), 1.49 (2H, m, H-11), 1.41 (1H, m, H-15α), 1.39 (1H, m, H-3β), 1.31 (1H, m, H-16β), 1.31 (1H, m, H-6β), 1.24 (1H, td, $J = 13.3$, 3.1 Hz, H-19α), 1.17 (1H, td, $J = 13.4$, 3.9 Hz, H-3α), 1.12 (3H, s, H-27), 1.07 (1H, dd, $J = 12.6$, 2.6 Hz, H-5), 1.03 (1H, t, $J = 3.4$ Hz, H-13), 1.02 (1H, dd, $J = 12.2$, 1.8 Hz, H-17), 0.96 (1H, td, $J = 13.0$, 3.9 Hz, H-1α), 0.93 (1H, m, H-12), 0.87 (3H, s, H-24), 0.83 (3H, s, H-29), 0.79 (3H, s, H-23), 0.79 (3H, s, H-28), 0.78 (3H, s, H-30), 0.65 (3H, s, H-25); for the NMR spectrum, see Supplementary Fig. 26.

### X-ray crystallographic analysis

Single crystals of compounds **5**, **7**, and **10** were grown in CH$_3$OH/CH$_2$Cl$_2$ (1:2, v/v), whereas those of **9** and **11** were grown in CH$_3$CN/CH$_2$Cl$_2$ (1:2, v/v), by a slow evaporation process at room temperature. Single crystal X-ray diffraction measurements were performed on a Bruker D8 Venture diffractometer using Cu Kα radiation at 213 K (for **5** and **7**), 203 K (for **9** and **11**), or 233 K (for **10**). The data collection was performed with the APEX3 program, and cell refinement and data reduction were carried out using the SAINT program. The structures of **5**, **7**, **9**, **10**, and **11** were solved by the direct method with the SHELXT program and refined using the SHELXL program. All non-hydrogen atoms were refined anisotropically, whereas hydrogen atoms were placed by geometrical calculations. The absolute configuration of **5**, **7**, **9**, **10**, and **11** was determined by the Flack parameters.

## Analysis of the metabolites produced by *A. homomorphus, A. fumigatus, A. alliaceus,* and *N. moseri* and the expression of biosynthetic genes

*A. homomorphus* CBS 101889, *A. fumigatus* CBS 144.89, and *A. alliaceus* CBS 536.65 were cultivated on five different agar plates, namely DPY agar, PDA (0.4% potato extract, 2% dextrose, 1.5% agar), MEA (1.7% malt extract, 0.3% mycological peptone, 2% agar), YES agar (2% yeast extract, 15% sucrose, 0.05% $MgSO_4 \cdot 7H_2O$, 0.001% $ZnSO_4 \cdot 7H_2O$, 0.0005% $CuSO_4 \cdot 5H_2O$, agar 2%, pH 6.4–6.6), and CZYA (0.3% $NaNO_3$, 0.1% $K_2HPO_4$, 0.05% KCl, 0.05% $MgSO_4 \cdot 7H_2O$, 0.001% $FeSO_4 \cdot 7H_2O$, 0.5% yeast extract, 3% sucrose, 0.001% $ZnSO_4 \cdot 7H_2O$, 0.0005% $CuSO_4 \cdot 5H_2O$, 1.5% agar, pH 6.0–6.5), and five different liquid media, namely DPY, PDB (0.4% potato extract, 2% dextrose), MEB (1.7% malt extract, 0.3% mycological peptone), YES (2% yeast extract, 15% sucrose, 0.05% $MgSO_4 \cdot 7H_2O$, 0.001% $ZnSO_4 \cdot 7H_2O$, 0.0005% $CuSO_4 \cdot 5H_2O$, pH 6.4–6.6), and CZY (0.3% $NaNO_3$, 0.1% $K_2HPO_4$, 0.05% KCl, 0.05% $MgSO_4 \cdot 7H_2O$, 0.001% $FeSO_4 \cdot 7H_2O$, 0.5% yeast extract, 3% sucrose, 0.001% $ZnSO_4 \cdot 7H_2O$, 0.0005% $CuSO_4 \cdot 5H_2O$, pH 6.0–6.5) for seven days at 25 °C. *N. moseri* CBS 164.80 was cultivated in the same manner as described above, but each medium was supplemented with 0.01% 4-HBA.

Metabolite extraction from the agar plates was performed as described above. When cultivated in the liquid medium, medium and mycelia were first separated by filtration. The mycelia were extracted with acetone with sonication for one hour, concentrated, and reextracted with ethyl acetate. The resultant crude extract was dissolved in methanol and analyzed by LC–MS.

For the analysis of triterpenoids, a solvent system of 20 mM formic acid (solvent A) and acetonitrile containing 20 mM formic acid (solvent B) was used, at a flow rate of 1.0 mL/min and a column temperature of 40 °C. Separation was performed using a linear gradient from 10:90 (solvent B/solvent A) to 70:30 for 2 min, 70:30 to 100:0 for the following 3 min, then 100:0 for the following 5 min, and a linear gradient from 100:0 to 10:90 within the following 1 min, and then 10:90 for 3 min of equilibrium. APCI source was used in positive polarity for mass spectrometry. The ion source temperature was set to 350 °C, and the curtain gas was 30 psi. For TOF-MS, the parameters were set as follows: nebulizer current 3 µA, CAD gas 8, declustering potential 80 V, declustering potential spread 0 V, collision energy 10 V, collision energy spread 0 V, TOF mass range 400–500 Da, and accumulation time 0.25 s. For TOF-MS/MS, the parameters were set as follows: nebulizer current 3 µA, CAD gas 8, declustering potential 80 V, declustering potential spread 0 V, collision energy 35 V, collision energy spread 15 V, TOF mass range 50–1000 Da, accumulation time 0.1 s.

For the analysis of sesquiterpene hydroquinones, a system of 20 mM formic acid (solvent A) and acetonitrile containing 20 mM formic acid (solvent B) was used, at a flow rate of 0.4 mL/min and a column temperature of 40 °C. Separation was performed using a linear gradient from 10:90 (solvent B/solvent A) to 100:0 for 10 min, 100:0 for the following 3 min, and a linear gradient from 100:0 to 10:90 within the following 2 min, and then 10:90 for 2.5 min of equilibrium. ESI source was used in negative polarity for mass spectrometry. The ion source temperature was set to 450 °C, and the curtain gas was 25 psi. For TOF-MS, the parameters were set as follows: spray voltage −4.5 kV, CAD gas 7, declustering potential −80 V, declustering potential spread 0 V, collision energy −10 V, collision energy spread 0 V, TOF mass range 320–340 Da, and accumulation time 0.25 s. For TOF-MS/MS, the parameters were set as follows: spray voltage −4.5 kV, CDS gas 7, declustering potential −80 V, declustering potential spread 0 V, collision energy −35 V, collision energy spread 0 V, TOF mass range 50–1000 Da, accumulation time 0.1 s.

Compounds **4, 11,** and **12** were detected in YES liquid medium, DPY liquid medium, and MEB, respectively. Mycelia of *A. homomorphus* CBS 101889, *A. alliaceus* CBS 536.65, and *N. moseri* CBS 164.80 from YES liquid medium, DPY liquid medium, and MEB, respectively, were collected, and subsequently, their total RNA was extracted using a GeneJET Plant RNA Purification Kit (Thermo Scientific). Complementary DNA (cDNA) synthesis was then performed with a HiScript III 1st Strand cDNA Synthesis Kit (+ gDNA wiper) (Vazyme Biotech). The expression of each gene was confirmed by PCR using the synthesized cDNA as a template (Supplementary Fig. 5).

### Antibacterial assay

Compounds **12** and **13** were tested for their antimicrobial activity using the Bauer-Kirby method. Five bacterial strains (*Staphylococcus epidermidis* ATCC 12228, *Staphylococcus aureus* ATCC 6538, *Bacillus cereus, Streptococcus faecalis,* and *Escherichia coli* ATCC 10536) were cultivated in Luria-Bertani (LB) broth at 37 °C for 12 h and then diluted to a concentration of ~$5 \times 10^5$ colony forming unit (CFU)/mL using Mueller Hinton broth (2 g/L beef infusion solids, 1.5 g/L starch, 17.5 g/L casein hydrolysate). Subsequently, 180 µL of the prepared microbial-containing medium was spread onto LB plates. After the surfaces of the plates dried, 5 mm filter papers impregnated with 5 µL of a continuous 2-fold dilution of compounds (from 10 to 1.25 mg/mL) were placed on each plate, and the LB plates were incubated at 37 °C. After 24 h, inhibition zones around each filter paper were measured. The experiment was performed in triplicate. Ampicillin was used as a positive control.

### Analytical data

**α-polypodatetraene (1).** Colorless oil; $[\alpha]^{22.2}_D + 28.6$ (*c* 0.94, $CHCl_3$); UV ($CHCl_3$) $\lambda_{max}$ (log ε) 200 (3.55) nm; IR (KBr) $\nu_{max}$ 2955, 2924, 2850, 1461, 1377 cm$^{-1}$; $^1$H NMR (600 MHz, $CDCl_3$): δ [ppm] 5.11 (m, 3H), 4.82 (q, *J* = 1.4 Hz, 1H), 4.54 (brd, *J* = 1.1 Hz, 1H), 2.39 (ddd, *J* = 12.8, 4.2, 2.5 Hz, 1H), 2.08 (m, 5H), 1.99 (m, 5H), 1.82 (m, 1H), 1.75 (m, 1H), 1.72 (m, 1H), 1.68 (s, 3H), 1.60 (s, 3H), 1.60 (s, 3H), 1.59 (m, 1H), 1.56 (s, 3H), 1.56 (m, 3H), 1.48 (m, 2H), 1.31 (qd, *J* = 13.0, 4.3 Hz, 1H), 1.17 (td, *J* = 13.4, 3.9 Hz, 1H), 1.08 (dd, *J* = 12.6, 2.7 Hz, 1H), 1.00 (td, *J* = 13.0, 3.9 Hz, 1H), 0.87 (s, 3H), 0.80 (s, 3H), 0.66 (s, 3H); $^{13}$C NMR (150 MHz, $CDCl_3$): δ [ppm] 148.8, 134.9, 134.9, 131.2, 125.1, 124.4, 124.3, 106.1, 56.2, 55.6, 42.2, 39.8, 39.7, 39.6, 39.1, 38.4, 33.6, 33.6, 26.9, 26.8, 26.7, 25.7, 24.5, 23.8, 21.7, 19.4, 17.7, 16.0, 16.0, 14.5; for NMR spectra, see Supplementary Figs. 12 and 13; HRMS (APCI) *m/z*: [M + H]$^+$ Calcd for $C_{30}H_{51}$ 411.3985; Found 411.3966. The NMR data are in good agreement with the reported data[38].

**α-polypodatetraen-3β-ol (2).** Colorless oil; $[\alpha]^{18.4}_D + 21.9$ (*c* 0.99, $CHCl_3$); UV ($CHCl_3$) $\lambda_{max}$ (log ε) 200 (3.58) nm; IR (KBr) $\nu_{max}$ 3446, 2956, 2926, 2852, 1461, 1378 cm$^{-1}$; $^1$H NMR (600 MHz, $CDCl_3$): δ [ppm] 5.11 (m, 3H), 4.84 (q, *J* = 1.4 Hz, 1H), 4.55 (brd, *J* = 0.9 Hz, 1H), 3.25 (dd, *J* = 11.8, 4.5 Hz, 1H), 2.40 (ddd, *J* = 12.8, 4.2, 2.5 Hz, 1H), 2.07 (m, 5H), 1.98 (m, 5H), 1.82 (m, 1H), 1.78 (dt, *J* = 13.2, 3.6 Hz, 1H), 1.73 (dddd, *J* = 12.9, 4.9, 2.5, 2.5 Hz, 1H), 1.69 (m, 1H), 1.68 (s, 3H), 1.60 (s, 3H), 1.60 (s, 3H), 1.58 (m, 2H), 1.56 (s, 3H), 1.41 (m, 4H), 1.15 (td, *J* = 13.3, 3.6 Hz, 1H), 1.07 (dd, *J* = 12.6, 2.7 Hz, 1H), 0.99 (s, 3H), 0.77 (s, 3H), 0.67 (s, 3H); $^{13}$C NMR (150 MHz, $CDCl_3$): δ [ppm] 148.1, 135.1, 134.9, 131.3, 124.9, 124.4, 124.2, 106.6, 78.9, 55.9, 54.6, 39.8, 39.7, 39.2, 39.1, 38.2, 37.1, 28.3, 27.9, 26.8, 26.7, 26.8, 25.7, 24.0, 23.9, 17.7, 16.0, 16.0, 15.4, 14.5; for NMR spectra, see Supplementary Figs. 14 and 15; HRMS (APCI) *m/z*: [M – $H_2O$ + H]$^+$ Calcd for $C_{30}H_{49}$ 409.3829; Found 409.3810. The NMR data are in good agreement with the reported data[39].

**8α-hydroxypolypoda-13,17,21-triene (3).** Colorless oil; $[\alpha]^{20.0}_D + 0.0$ (*c* 1.00, $CHCl_3$); UV ($CHCl_3$) $\lambda_{max}$ (log ε) 200 (3.59) nm; IR (KBr) $\nu_{max}$ 2955, 2925, 2852, 1460, 1378 cm$^{-1}$; $^1$H NMR (600 MHz, $CDCl_3$): δ [ppm] 5.18 (tq, *J* = 7.2, 1.1 Hz, 1H), 5.10 (m, 2H), 2.07 (m, 6H), 1.98 (m, 4H), 1.86 (dt, *J* = 12.4, 3.3 Hz, 1H), 1.68 (s, 3H), 1.64 (m, 1H), 1.61 (s, 3H), 1.60 (s, 3H), 1.60 (s, 3H), 1.57 (dt, *J* = 13.7, 3.7 Hz, 1H), 1.43 (m, 2H), 1.37 (m,

2H), 1.30 (m, 2H), 1.26 (m, 1H), 1.15 (td, $J$ = 13.5, 4.2 Hz, 1H), 1.13 (s, 3H), 1.03 (t, $J$ = 3.9 Hz, 1H), 0.98 (td, $J$ = 12.9, 3.6 Hz, 1H), 0.91 (dd, $J$ = 12.3, 2.4 Hz, 1H), 0.89 (m, 1H), 0.87 (s, 3H), 0.78 (s, 3H), 0.78 (s, 3H); $^{13}$C NMR (150 MHz, CDCl$_3$): δ [ppm] 135.2, 135.0, 131.2, 125.0, 124.4, 124.2, 74.1, 61.5, 56.2, 44.5, 42.0, 39.7, 39.7, 39.7, 39.1, 33.4, 33.2, 31.4, 26.8, 26.7, 25.7, 25.5, 23.8, 21.5, 20.6, 18.5, 17.7, 16.2, 16.0, 15.4; for NMR spectra, see Supplementary Figs. 16 and 17; HRMS (APCI) *m/z*: [M − H$_2$O + H]$^+$ Calcd for C$_{30}$H$_{51}$ 411.3985; Found 411.3966. The NMR data are in good agreement with the reported data[40].

**homomonoceroid A (4).** Colorless oil; $[\alpha]^{19.6}_D$ + 28.9 (*c* 0.45, CHCl$_3$); UV (CHCl$_3$) λ$_{max}$ (log ε) 205 (4.08) nm; IR (KBr) ν$_{max}$ 3420, 2938, 2870, 2846, 1459, 1388 cm$^{-1}$; for NMR data, see Supplementary Figs. 18–24; HRMS (APCI) *m/z*: [M − H$_2$O + H]$^+$ Calcd for C$_{30}$H$_{51}$O 427.3934; Found 427.3923.

**fumionoceroid A (5).** Colorless crystal; $[\alpha]^{18.7}_D$ + 14.5 (*c* 0.89, CHCl$_3$); UV (CHCl$_3$) λ$_{max}$ (log ε) 207 (4.68) nm; IR (KBr) ν$_{max}$ 3290, 2959, 2928, 2867, 1459, 1379 cm$^{-1}$; for NMR data, see Supplementary Figs. 27–33; HRMS (APCI) *m/z*: [M − H$_2$O + H]$^+$ Calcd for C$_{30}$H$_{49}$ 409.3829; Found 409.3811.

**fumionoceroid B (6).** White amorphous solid; $[\alpha]^{23.9}_D$ + 8.2 (*c* 0.69, CHCl$_3$); UV (CHCl$_3$) λ$_{max}$ (log ε) 207 (4.09) nm; IR (KBr) ν$_{max}$ 3394, 2957, 2927, 2853, 1459, 1379 cm$^{-1}$; for NMR data, see Supplementary Figs. 34–40; HRMS (APCI) *m/z*: [M − H$_2$O + H]$^+$ Calcd for C$_{30}$H$_{49}$O 425.3778; Found 425.3758.

**fumionoceroid C (7).** Colorless crystal; $[\alpha]^{19.9}_D$ + 18.1 (*c* 1.00, CHCl$_3$); UV (CHCl$_3$) λ$_{max}$ (log ε) 200 (3.60) nm; IR (KBr) ν$_{max}$ 3302, 2954, 2926, 2852, 1465, 1381 cm$^{-1}$; for NMR data, see Supplementary Figs. 41–47; HRMS (APCI) *m/z*: [M − H$_2$O + H]$^+$ Calcd for C$_{30}$H$_{49}$O 425.3778; Found 425.3762.

**alliaonoceroid A (8).** White amorphous solid; $[\alpha]^{23.2}_D$ + 5.3 (*c* 0.71, CHCl$_3$); UV (CHCl$_3$) λ$_{max}$ (log ε) 206 (4.08) nm; IR (KBr) ν$_{max}$ 3338, 2955, 2925, 2852, 1460, 1378 cm$^{-1}$; $^1$H NMR (600 MHz, C$_6$D$_6$): δ [ppm] 3.01 (m, 1H), 1.95 (tt, $J$ = 13.2, 3.2 Hz, 2H), 1.79 (td, $J$ = 13.1, 3.9 Hz, 1H), 1.75 (m, 2H), 1.69 (m, 1H), 1.63 (dt, $J$ = 12.7, 3.6 Hz, 1H), 1.59 (m, 1H), 1.56 (m, 1H), 1.51 (ddd, $J$ = 13.9, 6.1, 3.6 Hz, 1H), 1.46 (m, 2H), 1.41 (m, 1H), 1.46 (m, 3H), 1.37 (s, 3H), 1.36 (m, 3H), 1.35 (s, 3H), 1.25 (m, 2H), 1.16 (m, 3H), 0.96 (s, 3H), 0.86 (m, 1H), 0.86 (s, 3H), 0.83 (m, 1H), 0.80 (s, 3H), 0.75 (s, 3H), 0.75 (s, 3H), 0.74 (m, 1H), 0.69 (s, 3H); $^{13}$C NMR (150 MHz, C$_6$D$_6$): δ [ppm] 80.1, 79.9, 78.4, 61.1, 61.1, 56.5, 55.5, 45.8, 45.6, 42.3, 40.6, 39.1, 39.1, 38.9, 38.7, 33.7, 33.6, 28.4, 28.1, 25.7, 25.5, 25.5, 25.3, 21.7, 21.2, 20.7, 19.3, 16.1, 16.0, 15.7; for NMR spectra, see Supplementary Figs. 48 and 49; HRMS (APCI) *m/z*: [M − H$_2$O + H]$^+$ Calcd for C$_{30}$H$_{51}$O 427.3934; Found 427.3915. The NMR data are in good agreement with the reported data[42].

**alliaonoceroid B (9).** Colorless crystal; $[\alpha]^{23.5}_D$ + 48.2 (*c* 0.81, CHCl$_3$); UV (CHCl$_3$) λ$_{max}$ (log ε) 200 (3.61) nm; IR (KBr) ν$_{max\,3445}$, 2954, 2926, 2856, 1694, 1459, 1378 cm$^{-1}$; for NMR data, see Supplementary Figs. S50–S56; HRMS (APCI) *m/z*: [M − H$_2$O + H]$^+$ Calcd for C$_{30}$H$_{49}$O$_2$ 441.3727; Found 441.3715.

**alliaonoceroid C (10).** Colorless crystal; $[\alpha]^{23.0}_D$ + 15.5 (*c* 1.00, CHCl$_3$); UV (CHCl$_3$) λ$_{max}$ (log ε) 200 (3.64) nm; IR (KBr) ν$_{max}$ 2955, 2926, 2856, 1728, 1463, 1375, 1248 cm$^{-1}$; for NMR data, see Supplementary Figs. 57–63; HRMS (APCI) *m/z*: [M − H$_2$O + H]$^+$ Calcd for C$_{32}$H$_{53}$O$_2$ 469.4040; Found 469.4036.

**alliaonoceroid D (11).** Colorless crystal; $[\alpha]^{24.3}_D$ + 40.0 (*c* 1.00, CHCl$_3$); UV (CHCl$_3$) λ$_{max}$ (log ε) 200 (3.67) nm; IR (KBr) ν$_{max}$ 2958, 2926, 2871, 1737, 1462, 1367, 1244 cm$^{-1}$; for NMR data, see Supplementary Figs. 64–70; HRMS (APCI) *m/z*: [M − AcOH + H]$^+$ Calcd for C$_{32}$H$_{53}$O$_3$ 485.3989; Found 485.3989.

**moserinol (12).** White amorphous solid; $[\alpha]^{22.8}_D$ + 2.1 (*c* 1.00, CHCl$_3$); UV (CHCl$_3$) λ$_{max}$ (log ε) 215 (3.54), 233 (3.41), 294 (3.54) nm; IR (KBr) ν$_{max}$ 3391, 2925, 1505, 1459, 1388, 1200 cm$^{-1}$; for NMR data, see Supplementary Figs. 71–77; HRMS (ESI) *m/z*: [M − H]$^-$ Calcd for C$_{21}$H$_{31}$O$_3$ 331.2279; Found 331.2283.

***ent*-yahazunol (13).** White amorphous solid; $[\alpha]^{21.8}_D$ + 7.3 (*c* 0.10, CHCl$_3$); UV (CHCl$_3$) λ$_{max}$ (log ε) 206 (3.33), 215 (3.37), 237 (3.24), 294 (3.25) nm; IR (KBr) ν$_{max}$ 3382, 2935, 1506, 1458, 1392, 1234 cm$^{-1}$; $^1$H NMR (600 MHz, acetone-$d_6$): δ [ppm] 6.64 (d, $J$ = 2.8 Hz, 1H), 6.52 (d, $J$ = 8.5 Hz, 1H), 6.49 (dd, $J$ = 8.5, 2.8 Hz, 1H), 2.85 (dd, $J$ = 14.9, 2.0 Hz, 1H), 2.39 (dd, $J$ = 15.0, 6.1 Hz, 1H), 1.92 (dt, $J$ = 12.4, 3.2 Hz, 1H), 1.83 (brd, $J$ = 13.0 Hz, 1H), 1.67 (m, 1H), 1.63 (m, 1H), 1.58 (td, $J$ = 14.8, 3.9 Hz, 1H), 1.58 (dd, $J$ = 5.9, 2.0 Hz, 1H), 1.36 (m, 1H), 1.34 (m, 1H), 1.32 (m, 1H), 1.30 (s, 3H), 1.10 (td, $J$ = 13.5, 4.2 Hz, 1H), 0.97 (s, 3H), 0.96 (dd, $J$ = 12.3, 2.6 Hz, 1H), 0.86 (s, 3H), 0.82 (s, 3H), 0.72 (td, $J$ = 13.2, 3.5, 1H); $^{13}$C NMR (150 MHz, acetone-$d_6$): δ [ppm] 150.3, 149.7, 131.0, 118.8, 117.4, 114.2, 75.0, 62.2, 56.9, 44.4, 42.4, 41.3, 40.5, 33.7, 33.7, 27.8, 24.5, 21.8, 21.1, 19.0, 15.8; for NMR spectra, see Supplementary Figs. 78 and 79; HRMS (ESI) *m/z*: [M − H]$^-$ Calcd for C$_{21}$H$_{31}$O$_3$ 331.2279; Found 331.2288. The NMR data are in good agreement with the reported data for yahazunol[47], and the absolute configuration was determined by comparing the specific rotation of **13** with that reported for *ent*-yahazunol[46].

**Crystallographic data for 5.** C$_{30}$H$_{50}$O, $M$ = 426.70, $a$ = 19.8821(4) Å, $b$ = 19.8821(4) Å, $c$ = 7.7762(2) Å, $\alpha$ = 90°, $\beta$ = 90°, $\gamma$ = 90°, $V$ = 3073.92(15) Å$^3$, $T$ = 213(2) K, space group P4$_3$, $Z$ = 4, $\mu$(Cu Kα) = 0.395 mm$^{-1}$, 38 497 reflections measured, 6229 independent reflections ($R_{int}$ = 0.0575). The final $R_1$ values were 0.0403 ($I$ > 2σ(*I*)). The final $wR(F^2)$ values were 0.1037 ($I$ > 2σ(*I*)). The final $R_1$ values were 0.0413 (all data). The final $wR(F^2)$ values were 0.1054 (all data). The goodness of fit on $F^2$ was 1.034. Flack parameter 0.03(8). The crystallographic information file (CIF) for this crystal structure was submitted to The Cambridge Crystallographic Data Centre (CCDC) under reference number 2279439.

**Crystallographic data for 7.** C$_{30.5}$H$_{52}$O$_{2.5}$, $M$ = 458.72, $a$ = 12.7561(3) Å, $b$ = 25.8142(7) Å, $c$ = 8.2201(2) Å, $\alpha$ = 90°, $\beta$ = 90°, $\gamma$ = 90°, $V$ = 2706.78(12) Å$^3$, $T$ = 213(2) K, space group P2$_1$2$_1$2, $Z$ = 4, $\mu$(Cu Kα) = 0.523 mm$^{-1}$, 36 412 reflections measured, 5545 independent reflections ($R_{int}$ = 0.1262). The final $R_1$ values were 0.0536 ($I$ > 2σ(*I*)). The final $wR(F^2)$ values were 0.1306 ($I$ > 2σ(*I*)). The final $R_1$ values were 0.0684 (all data). The final $wR(F^2)$ values were 0.1411 (all data). The goodness of fit on $F^2$ was 1.054. Flack parameter 0.1(2). The crystallographic information file (CIF) for this crystal structure was submitted to The Cambridge Crystallographic Data Centre (CCDC) under reference number 2279440.

**Crystallographic data for 9.** C$_{30}$H$_{50}$O$_3$, $M$ = 458.70, $a$ = 6.3025(2) Å, $b$ = 25.5737(8) Å, $c$ = 16.2273(5) Å, $\alpha$ = 90°, $\beta$ = 99.296(2)°, $\gamma$ = 90°, $V$ = 2581.14(14)) Å$^3$, $T$ = 203(2) K, space group P2$_1$, $Z$ = 4, $\mu$(Cu Kα) = 0.564 mm$^{-1}$, 43 044 reflections measured, 10 447 independent reflections ($R_{int}$ = 0.0974). The final $R_1$ values were 0.0544 ($I$ > 2σ(*I*)). The final $wR(F^2)$ values were 0.1293 ($I$ > 2σ(*I*)). The final $R_1$ values were 0.0587 (all data). The final $wR(F^2)$ values were 0.1325 (all data). The goodness of fit on $F^2$ was 1.055. Flack parameter −0.10(12). The crystallographic information file (CIF) for this crystal structure was submitted to The Cambridge Crystallographic Data Centre (CCDC) under reference number 2279441.

**Crystallographic data for 10**. $C_{30.5}H_{52}O_{2.5}$, $M = 486.75$, $a = 11.6771(3)$ Å, $b = 7.3892(2)$ Å, $c = 33.1184(8)$ Å, $\alpha = 90°$, $\beta = 93.8920(10)°$, $\gamma = 90°$, $V = 2851.01(13)$ Å$^3$, $T = 233(2)$ K, space group $P2_1$, $Z = 4$, $\mu$(Cu K$\alpha$) = 0.537 mm$^{-1}$, 43 244 reflections measured, 10785 independent reflections ($R_{int} = 0.0437$). The final $R_1$ values were 0.0403 ($I > 2\sigma(I)$). The final $wR(F^2)$ values were 0.1176 ($I > 2\sigma(I)$). The final $R_1$ values were 0.0417 (all data). The final $wR(F^2)$ values were 0.1197 (all data). The goodness of fit on $F^2$ was 1.022. Flack parameter −0.01(6). The crystallographic information file (CIF) for this crystal structure was submitted to The Cambridge Crystallographic Data Centre (CCDC) under reference number 2279442.

**Crystallographic data for 11**. $C_{34}H_{56}O_5$, $M = 544.78$, $a = 7.8398(3)$ Å, $b = 10.0596(4)$ Å, $c = 10.6376(4)$ Å, $\alpha = 74.8350(10)°$, $\beta = 75.3740(10)°$, $\gamma = 83.1520(10)°$, $V = 782.21(5)$ Å$^3$, $T = 203(2)$ K, space group $P1$, $Z = 1$, $\mu$(Cu K$\alpha$) = 0.590 mm$^{-1}$, 20 283 reflections measured, 6036 independent reflections ($R_{int} = 0.0496$). The final $R_1$ values were 0.0415 ($I > 2\sigma(I)$). The final $wR(F^2)$ values were 0.1142 ($I > 2\sigma(I)$). The final $R_1$ values were 0.0420 (all data). The final $wR(F^2)$ values were 0.1150 (all data). The goodness of fit on $F^2$ was 1.096. Flack parameter 0.17(8). The crystallographic information file (CIF) for this crystal structure was submitted to The Cambridge Crystallographic Data Centre (CCDC) under reference number 2279443.

### Reporting summary
Further information on research design is available in the Nature Portfolio Reporting Summary linked to this article.

## Data availability
The manually curated fungal BGCs and custom-made HMM profiles are deposited at Zenodo under the https://doi.org/10.5281/zenodo.8126803[82]. The online version of the FunBGCs database can currently be accessed at http://staffweb1.cityu.edu.hk/ymatsuda/funbgcs/funbgcs.html. The corresponding genomic regions for the biosynthetic gene clusters reported in this study can be accessed at the National Center for Biotechnology Information (NCBI) database with the accession numbers NW_020291903 [https://www.ncbi.nlm.nih.gov/nuccore/NW_020291903.1?report=genbank&from=95123&to=101061&strand=true] (the *homo* cluster; region: 95,123…101,061), DS499601 (the *fumi* cluster; region: 75,155…91,115), NW_022474693 [https://www.ncbi.nlm.nih.gov/nuccore/NW_022474693.1?report=genbank&from=128036&to=138909] (the *alli* cluster; region: 128,036…138,909), and NW_026055164 [https://www.ncbi.nlm.nih.gov/nuccore/NW_026055164.1?report=genbank&from=259950&to=270410] (the *mos* cluster; region: 259,950…270,410). The crystallographic data obtained in this study have been deposited at the Cambridge Crystallographic Data Centre (CCDC) under deposition numbers 2279439 (**5**), 2279440 (**7**), 2279441 (**9**), 2279442 (**10**), and 2279443 (**11**). Copies of the crystallographic data can be obtained free of charge via https://www.ccdc.cam.ac.uk/structures/. The raw NMR data obtained in this study have been deposited at the Natural Product Magnetic Resonance Database (NP-MRD) under deposition numbers NP0332710 (**1**), NP0332711 (**2**), NP0332712 (**3**), NP0332713 (**4**), NP0332714 (**5**), NP0332715 (**6**), NP0332716 (**7**), NP0332717 (**8**), NP0332718 (**9**), NP0332719 (**10**), NP0332720 (**11**), NP0332721 (**12**), and NP0332722 (**13**). Copies of the NMR data can be obtained free of charge via https://np-mrd.org/. Source data are provided with this paper.

## Code availability
All original codes for FunBGCeX are deposited at Zenodo under the https://doi.org/10.5281/zenodo.8126797[83] and are also available at https://github.com/ydmatsd/funbgcex[84].

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

## Acknowledgements

We thank Prof. Katsuya Gomi (Tohoku University) and Profs. Katsuhiko Kitamoto and Jun-ichi Maruyama (University of Tokyo) for the expression vectors and fungal strain. We are also grateful to Dr. Man Kit Tse and Dr. Kwok Chung Law (City University of Hong Kong) for the NMR spectra acquisition and Dr. Shek Man Yiu (City University of Hong Kong) for X-ray diffraction data collection and analysis. This work was supported by General Research Fund grants from the Research Grants Council of Hong Kong (Project Nos. 11301321 and 11309022 to Y.M.).

## Author contributions

Y.M. designed the research and conducted the bioinformatic analysis. J.T. performed experiments. Both authors analyzed the data and co-wrote the manuscript.

## Competing interests

The authors declare no competing interests.
