## [Peer Review File · Nature Communications]

Discovery of fungal onoceroid triterpenoids through domainless enzyme-targeted global genome miningREVIEWER COMMENTS

Reviewer #1 (Remarks to the Author):

Tang and Matsuda have developed a new genome mining tool that was applied in a very careful genome analysis, resulting in the identification of several interesting fungal biosynthetic gene clusters with a Pyr4 homolog. Three of these gene clusters were selected for further study, leading to the discovery of several onoceroids, a class of natural products that was unknown from fungi before. This is also the first study in which Pyr4 cyclases only act on squalene and not in the biosynthesis of meroterpenoids. The manuscript contains a large body of highest quality results that are very accurately described. There is absolutely nothing to criticize. Only two small points:

1. Is "domainless biosynthetic enzymes" the correct term? Or are the domains just different to what is known and cannot easily be found?
2. line 84: Please explain the rationale for the "size of the gene product". And which sizes were included / excluded?

Reviewer #2 (Remarks to the Author):

Genomes of most microorganisms including fungi encode a huge number of biosynthesis gene clusters for natural products (BGCs), many of which have found important functions as antibiotics like penicillin or antitumor therapeutics. Most of the encoded compounds are unknown until today because the majority of BGCs is silent when microorganisms are cultivated under laboratory conditions, thus the produced compounds cannot be identified. In recent years, several bioinformatic tools have been improved to predict BGCs in genomes of microorganisms and in particular to identify BGCs potentially producing novel compounds. In the current manuscript, the authors now developed a fungal genome mining tool, which was based on a manually curated fungal BGC database and custom-made HMM profiles. By cloning of identified BGCs and their expression in a heterologous fungal host, the authors demonstrated that their genome mining tool could identify BGCs for previously in fungi undiscovered natural products, i.e., onoceroids that were previously identified in bacteria and are also present in animals and plants. Thus, the authors succeeded in the identification of the first onoceroid synthase in fungi.

Major comments

The manuscript is very well written, it was a pleasure to read it. The methods applied are state-of-the-art. Also, I think the authors tackled an important question, i.e., how can we get hold on the treasure of natural products in nature. In a recent paper by Y. Tang (cited by the authors), it was commented that our current genome mining tools mainly target "known-unknown" BGCs which produce unknown natural products synthesized by the known classes of core enzymes. Recent studies have highlighted the importance of genome mining-driven discovery of "unknown-unknown" natural products. However, also in this manuscript the authors analyzed „unknown-known“, i.e., potentially unknown enzymes producing known products. Thus, the main question is not solved yet, how can we discover non-canonical BGCs?

Specific critical comments:

1. As the authors rightly argued: The onoceroid BGCs characterized in this study could have also been identified using available BGC identification programs like antiSMASH. The advantage of the authors' approach is that apparently the BGCs extracted by antiSMASH contained substantially more genes outside the boundary of a BGC than those extracted by their method. Also, antiSMASH could apparently not detect the presence of pyr4-like genes in these BGCs. However, again, these BGCs could be discovered by standard BLAST search using Pyr4 as a query and subsequent manual investigation of the flanking regions of each identified pyr4-like gene, which, however, as the authors correctly argued, requires tedious procedures and might take weeks to months.
2. Also, onoceroids have been previously isolated from bacteria, ferns, higher plants, and animals, so no real new compound family was discovered but new derivatives. What was discovered is, onoceroids in fungi.
3. The authors should provide at least some data on potential bioactivity of the fungal onoceroid

derivatives.

4. Also, the authors state that „fungal onoceroid BGCs might be silent under standard laboratory conditions, which could explain why onoceroids had never been isolated from fungi.“ To lend weight to their discovery strategy it would be reasonable to measure the expression of a respective BGC at least in one of the identified fungi by RT-PCR or Northern blot.

Minor comments

1. Line 30: Please also cite Bergmann et al. 2007, Nat Chem Biol, who published the first example of expression of a silent gene cluster in fungi.

2. Line 50: "...selective extraction..." Please clarify, what do you mean? Selective extraction of BGCs for a distinct purpose?

3. line 116: a non-reference strain is misleading, there is no accepted reference strain. Maybe you meant a non-sequenced strain?

4. line 126: Any idea why "most of the Pyr4-like proteins identified (1010/1050) are of ascomycete origin"?

5. Please provide some more data on the recombinant strains of *Aspergillus*. Did you check how many copies of the plasmids were integrated into the genome? How did you proof that the strains contained the full plasmid?

Reviewer #3 (Remarks to the Author):

Comments for Author

Dr. Jia Tang and Dr. Yudai Matsuda's study, titled 'Discovery of fungal onoceroid triterpenoids through domainless enzyme-targeted global genome mining,' introduces a new genome mining pipeline termed FunBGCeX (Fungal BGC eXtractor). The authors have performed heterologous expression experiments in *Aspergillus oryzae* using three homologous *Aspergillus* BGCs identified by FunBGCeX. Subsequent chemical validation of the resultant compounds was performed using GC-MS, HPLC, HRMS, NMR, and X-ray crystallography. Commendably, the authors have provided open access to the data generated by their FunBGCeX program through a well-documented GitHub repository, <https://github.com/ydmatsd/funbgcex>. In addition, they have established a user-friendly GUI website, accessible at <http://staffweb1.cityu.edu.hk/ymatsuda/funbgcs/funbgcs.html>, which is particularly beneficial for users with limited computational skills. The creation of this GUI website is a commendable effort to enhance the program's accessibility. Exploration of the interface revealed it to be highly intuitive, with useful several features, such as the integrated BLASTp functionality and the capability to copy the amino acid sequences of any protein. However, as detailed below, it is questionable if this works adds to the field of genome mining.

Comment 1 (Genome Mining): The authors present two major limitations in the leading fungal genome mining program:

Lines 40-41: "First, antiSMASH classifies many genes as "other genes," even when they encode a biosynthetic enzyme."

Lines 49-52: "Furthermore, although antiSMASH and the other aforementioned tools can be used to extract all possible BGCs from a genome sequence, they do not facilitate the selective extraction of BGCs. Thus, users need to select BGCs based on the criteria set for subsequent wet lab experiments."

As I understand their critiques, both could have been solved from a post-processing program/software as they revolve around how the antiSMASH predictions are presented to the user (e.g., domains, category of gene, ability to filter), and not the underlying predictions themselves. As an example, one could have made a program that parses the output of antiSMASH and add in additional labels for mislabeled "other genes" while adding in options for the users to easily filter the antiSMASH BGC predictions by its presence of certain homologous genes, domains, or backbone classifications.

Given these two presented critiques, I struggle to see the need for an entirely new genome mining

platform. The authors must better clarify what separates their new genome mining program from existing options? Additionally, as the authors opted to create an entirely new genome mining program it is important for them to sufficiently indicate how their program performs better or differently from existing genome mining alternative. On this point, I will indicate several lines of concern that I have:

On lines 56-75 the authors created manually curated HMMER models for "domainless" proteins (as defined by it not having a detectable domain in the pFam database). From my rough estimation of Table S1 and S2 this led to the addition of between 50-100 previously unlabeled domains. The authors then created a set of rules based on the presence of biosynthetic domains labeled from their curated training database of already characterized clusters to determine the window size of these BGC predictions (line 76-88). It is important to note that while their addition of these 50-100 unlabeled domains is novel, the idea of using author specified rules to determine BGC boundaries is analogous to how many existing fungal genome mining programs have been created (including antiSMASH, SMURF).

The authors present 1,050 BGCs that contained a homologous copy of the Pyr4 gene (lines 100-102). However, the authors fail to ever present a comparison between their algorithm and antiSMASH or other competitors. How many of these BGCs would have similarly been identified by antiSMASH, even if the Pyr4 gene is labeled as "Other Gene"? As the authors have not presented such data, it is difficult to make the case for using their program as opposed to existing solutions within the natural products community.

The authors did validate three different Pyr4 containing proteins using heterologous expression (the chemistry and molecular biology in this paper will be addressed later in the review). However, as they state on (lines 209-211) "The onoceroid BGCs characterized in this study could also be identified using antiSMASH; ..." They counter this by stating that antiSMASH included more genes in the predictions than FunBGCeX (lines 210-211), however failed to present any data suggesting that the additional genes included by antiSMASH aren't a part of the native BGC. Were these additional antiSMASH genes heterologously expressed? If they were not, what evidence do the authors have to suggest they are not involved in the production of the natural product in the native organism?

The authors own admission that their validated BGCs could have been identified by the antiSMASH, in which the paper presents itself as a competitor, is concerning. If FunBGCeX is unable to prove that it can predict novel or different BGCs than antiSMASH I doubt it will have large adoption by the fungal natural product community. It also stands in direct contradiction with the line 215 where they state "the onoceroid BGCs would not have been effectively discovered using antiSMASH or conventional genome mining methodologies." As the onoceroid BGC is detectable by antiSMASH, I fail to see how this statement is true.

To summarize my concerns, I feel the authors have not shown any data to suggest their program performs any better than existing genome mining algorithms. Their validated BGCs could have been identified by antiSMASH and they utilized the same underlying framework that early iterations of antiSMASH used (i.e., rule based BGC calling). In addition, as the authors used a database of only characterized BGCs, their program falls under the same issue as many existing genome mining algorithms to being biased to only finding BGC architectures like what is already characterized.

Comment 2) Chemical analysis

The authors performed the structural analysis of eleven compounds including several new derivatives of onoceroid. Isolation, purification, and structure determination and elucidation of compounds were described well. I have some curiosities on the discussion and may ask that requires more details on the chemical analysis.

1. From Figure 3b, there is a small peak presented on the spectrum of NSAR1 extract, which is a control of this study. The peak disappeared on some of the mutants and also increased on others. The heterologous expressed genes may affect the chemical composition of the original primary/secondary metabolites of *A. oryzae*. In this case, I would recommend the authors to provide squalene and/or oxidosqualene production differences between control strain and mutants

through LC-MS or GC-MS since the proposed biosynthetic pathway of new compounds shared the precursor squalene from the host strain. Squalene can be used to produce ergosterol or lanosterol.

2. From Figure 3c, it would be better to indicate new compounds with different colored numbers. For example, write numbers of known compounds in black while write numbers of new compounds in blue. It may be easier to recognize the similarities and differences between known and new compounds for the authors.

3. From the discussion section (line 236), the authors mentioned that "onoceroids have never been isolated from fungi. Thus, the present study provides the first examples of fungal onoceroids and their biosynthetic pathways." Yes, onoceroids have never been isolated from fungi; however, onocerane carbon backbone was detected and isolated from fungi. Although this is first time the author isolated the onoceroids from heterologous expressed fungal genes, it would be hard to agree that this is the first examples of true fungal onoceroids since the compounds were not detected from the original fungal strains. It may be true that the compounds were derived from the fungal genome, but they are not natural products yet until they are isolated or detected from genetically unmodified fungal strains.

4. From scheme 1, intermediate compounds 12 to 17 were not detected or identified. In this figure and the last paragraph of the discussion, it sounds like that they are the actual compounds. I would recommend removing the intermediates or clearly mentioned them they are theoretical intermediate compounds on the writing. Also, square brackets are recommended for the intermediates of biosynthetic pathway from the scheme 1.

5. For the new compounds, UVmax and FT-IR data are necessary.

6. For the optical rotation, why is it recorded on different conditions? Usually, people use 23 or 20. The temperature should be the same. According to the references that the authors made, they were recorded under 23.

7. Please provide combined ¹H table and ¹³C table for the compounds with the positions. In this case, it would be better to detect the differences easily (especially for the carbons). I understand that the authors already provided the separated tables for individual compounds from the supplementary information. However, it is still recommended.

8. For the new compounds with overlaps such as compound 4, enlarged 2-D NMR of the overlapped section with correct indication is required. Otherwise, the authors may provide the raw NMR files for the supplementary data.

9. For compound 4, is there a specific reason to take NMR with C₆D₆ instead of CDCl₃? As the authors already know, the NMR of most of the known onoceroids were identified and determined with CDCl₃. If there is no specific reason, it would be better to change the NMR data for compound 4 to CDCl₃ one.

Comment 3) Biosynthetic pathway

1. While the onocenoids have prior not been detected in Fungi, the onocerane backbone, which is the same carbon backbone of onocenoids and upper chemical class of the compounds, have been characterized and found across many fungal taxa. At the end of the introduction (line 51), the authors emphasized obtaining a new class of metabolites through their genome mining tool. However, as there are several known biosynthetic pathways of the onocerane backbone in bacteria and plants, onocenoids cannot be correctly classified as a new class of natural products. To claim a new class of metabolites, the compounds should have either different carbon backbone or different backbone attachment instead of adding a single hydroxyl group or an acetyl group.

2. From line 58, the authors mentioned that "these domainless biosynthetic enzymes often drive diverse chemical reactions." A strong reference to support this claim must be included here. Some examples of domainless enzymes serving biosynthetic roles is not enough evidence that they are "often" biosynthetic as opposed to being a gap gene, pseudo gene, or having some alternative function. I would recommend removing the word "often".

3. From line 105, a PaxC-like prenyltransferase is specifically required for the biosynthesis of indole sesquiterpenoids or diterpenoids. The given reference 29, only showed the example of indole diterpenoids. The authors may add and edit correct references.

4. From scheme 1, is it also possible that oxidosqualene is produced first and converted to the final products? When I read provided references of the biosynthetic pathways from bacteria and plants, it may also be possible to be converted from the oxidosqualene.

5. The authors should provide the detailed methods and results on the strain construction. To describe the biological results of the strain construction the following questions should be answered thoroughly:

1. Please provide the raw data and methodology showing the validation of the fungal transformation plasmids.
2. Was the expression of the genes on the fungal transformation plasmids in *Aspergillus oryzae* validated (qPCR, semiQ, etc.). Please provide the data if so.
6. It appears that Figure 1S is nearly identical to Figure 1 in Matsuda's 2018 Nature Communications. To avoid any issues with self-plagiarism this figure should be re-drawn.

Reviewer #4 (Remarks to the Author):

Matsuda et al reported the discovery of a series of onoceroid compounds encoded by fungal biosynthetic gene clusters (BGCs). Those compounds were not isolated from the producing strain but generated through in vivo co-expression of different combinations of biosynthetic genes from the putative BGCs. A rigorous structural characterization was carried out by NMR, MS, Mosher's method and X-ray crystallography. Authors also described in detail a genome mining tool developed for fungal BGCs that incorporates a manually curated in-house database. Using this tool, authors input Pyr4-family terpene cyclases as the target gene, and found a number of BGCs that may encode novel terpene structures. Although this work is interesting, the overall significance of the genome mining method, and the significance of the function of Pyr4-family terpene cyclase is in question. There are several aspects mentioned below as suggestions to improve this manuscript. If significant improvements cannot be made, it is suggested that this work is too preliminary for Nat. Comm. and may be more suitable for more specialized journals.

1. Development of a fungal genome mining tool

Since this tool contains a manually curated database, it is expected to be more accurate in gene annotation for natural product biosynthetic enzymes. Except that, it seems that current genome mining tools (such as antiSMASH) can achieve all the functions claimed for this tool. In addition, there seems to be a lot of manual manipulation of data when using this tool, which does not sound user-friendly comparing to antiSMASH. In this manuscript, authors only used this tool to discover Pyr4-like protein containing BGCs. Perhaps authors could consider discover more types of BGCs using different proteins to show the broad applicability of this tool?

2. Function of Pyr4-family terpene cyclase

In this manuscript, authors only used heterologous host to express different combinations of biosynthetic proteins, identified corresponding products, and used those data to predict the biosynthetic path and claimed the terpene cyclase-catalyzed reaction. To more rigorously characterize this novel cyclase, the enzyme should be studied in more detail. The true substrate and any possible cofactors need to be incubated with the enzyme in vitro, and product structures need to be characterized. Mechanistic studies by mutagenesis and co-crystallization can also add more significance of this study.

3. Gene cluster identification and expression

The characterized structures are carbon skeleton with minimum oxidation. Authors also stated that compound 8 is a biosynthetic precursor of a natural product. Considering the fact that they are heterologously expressed using at most five proteins, is it possible that this type of BGC have greater potential for structural diversity? Wild-type strains containing this type of BGC can be isolated to find more structural analogs. In addition, Fig 2 panel b showed a couple of such BGCs contain "other" proteins. Heterologous expression of the whole cluster (not just selected proteins) with "other" genes can also be attempted.

4. Why did authors conduct research on fungal onoceroid triterpenoids

If the authors can explore interesting medical uses or any other applications of those products, the novelty of this manuscript can be improved.

Responses to the reviewers' comments

Reviewer #1

Tang and Matsuda have developed a new genome mining tool that was applied in a very careful genome analysis, resulting in the identification of several interesting fungal biosynthetic gene clusters with a Pyr4 homolog. Three of these gene clusters were selected for further study, leading to the discovery of several onocerooids, a class of natural products that was unknown from fungi before. This is also the first study in which Pyr4 cyclases only act on squalene and not in the biosynthesis of meroterpenoids. The manuscript contains a large body of highest quality results that are very accurately described. There is absolutely nothing to criticize. Only two small points:
RESPONSE: We greatly appreciate the reviewer's favorable comments and valuable suggestions on our manuscript. Please see our responses to each question below.

1. Is "domainless biosynthetic enzymes" the correct term? Or are the domains just different to what is known and cannot easily be found?

RESPONSE: In this manuscript, we defined domainless proteins as proteins that currently lack a detectable protein domain, as we could not find a suitable term used to describe such proteins. We mention this in the manuscript as follows: "...enzymes without a detectable protein domain (hereafter termed "domainless enzymes")..."

2. line 84: Please explain the rationale for the "size of the gene product". And which sizes were included / excluded?

RESPONSE: We considered the size of a gene product, as genome sequence data sometimes contain small proteins that are apparently mispredicted and nonfunctional. To make the analytic procedure simple, we ignored small proteins that do not seem to be a biosynthetic protein. To clarify this point, we have changed "the size of the gene product" to "whether the gene encoded a small protein (to ignore mispredicted small proteins)." The protein size to be considered small can be set by the user, but as a default, we set the size as 200 amino acid residues. The details are given in the Methods section as follows: "Small proteins with < 200 amino acid residues were removed from the subsequent analysis unless they were classified as biosynthetic proteins."

Reviewer #2

Genomes of most microorganisms including fungi encode a huge number of biosynthesis gene clusters for natural products (BGCs), many of which have found important functions as antibiotics like penicillin or antitumor therapeutics. Most of the encoded compounds are unknown until today because the majority of BGCs is silent when microorganisms are cultivated under laboratory

conditions, thus the produced compounds cannot be identified. In recent years, several bioinformatic tools have been improved to predict BGCs in genomes of microorganisms and in particular to identify BGCs potentially producing novel compounds.

In the current manuscript, the authors now developed a fungal genome mining tool, which was based on a manually curated fungal BGC database and custom-made HMM profiles. By cloning of identified BGCs and their expression in a heterologous fungal host, the authors demonstrated that their genome mining tool could identify BGCs for previously in fungi undiscovered natural products, i.e., onoceroids that were previously identified in bacteria and are also present in animals and plants. Thus, the authors succeeded in the identification of the first onoceroid synthase in fungi.

RESPONSE: We greatly appreciate the reviewer's time and constructive comments and suggestions on our manuscript. We have revised our manuscript according to the reviewer's comments, and we believe that our manuscript has been improved. Please see below our responses to each comment/question.

Major comments

The manuscript is very well written, it was a pleasure to read it. The methods applied are state-of-the-art. Also, I think the authors tackled an important question, i.e., how can we get hold on the treasure of natural products in nature. In a recent paper by Y. Tang (cited by the authors), it was commented that our current genome mining tools mainly target "known-unknown" BGCs which produce unknown natural products synthesized by the known classes of core enzymes. Recent studies have highlighted the importance of genome mining-driven discovery of "unknown-unknown" natural products. However, also in this manuscript the authors analyzed „unknown-known“, i.e., potentially unknown enzymes producing known products. Thus, the main question is not solved yet, how can we discover non-canonical BGCs?

RESPONSE: We understand this is an important issue in genome mining and that our present methodology cannot fully address this challenge. Nevertheless, we believe that our present study has provided a partial solution to discover "unknown-unknown" BGCs, as several BGCs that cannot be detected by antiSMASH were extracted with our genome mining tool (BGCs indicated by red asterisks in Figure 2b). These BGCs lack well-known core enzyme genes, but as we now consider Pyr4-like proteins as core enzymes, our method allows for the extraction of such BGCs without a typical core enzyme gene. In addition, with many more types of enzymes other than Pyr4-family terpene cyclases being classified as core enzymes, our method can detect a number of BGCs that are overlooked by other genome mining tools. To emphasize the advantageous point of our approach, we have heterologously expressed one of the BGCs not detected by antiSMASH and obtained a new sesquiterpene hydroquinone named moseranol. Although moseranol lacks a

structural novelty, our study has provided a new form of biosynthetic mechanism with a Pyr4 homologue. The manuscript has now been updated with these additional experimental results. Please note that we did not investigate the biosynthetic pathway of moserinol, as it is not the main focus of our study.

Furthermore, as mentioned in the main text, we are now improving our genome mining tool and adding more functions, including the extraction of BGCs lacking a known core enzyme gene. Such “coreless” BGCs can be detected by focusing on one or more types of biosynthetic genes and extracting genomic regions around these genes. We hope to publish another work with the improved genome mining method in the future.

Collectively, although we understand that there are some limitations in our present method, we are still confident in the usefulness of our method in discovering BGCs that are difficult to be detected by other available tools.

Specific critical comments:

1. As the authors rightly argued: The onoceroid BGCs characterized in this study could have also been identified using available BGC identification programs like antiSMASH. The advantage of the authors' approach is that apparently the BGCs extracted by antiSMASH contained substantially more genes outside the boundary of a BGC than those extracted by their method. Also, antiSMASH could apparently not detect the presence of pyr4-like genes in these BGCs. However, again, these BGCs could be discovered by standard BLAST search using Pyr4 as a query and subsequent manual investigation of the flanking regions of each identified pyr4-like gene, which, however, as the authors correctly argued, requires tedious procedures and might take weeks to months.

RESPONSE: As also asked by the other reviewers, we have analyzed the same set of ~2,000 fungal genomes using antiSMASH and added one paragraph in the Results section to describe the analysis result (lines 147–158). First, we realized that the analysis of 2,000 genomes by antiSMASH required a long time (approximately two weeks with a standard laptop), as well as manual inspection of each extracted BGC to identify those with a desired feature. In addition, antiSMASH was unable to detect some of the BGCs discovered by our tool. Thus, we believe that our approach is advantageous when one would like to focus on BGCs with a specific feature.

2. Also, onoceroids have been previously isolated from bacteria, ferns, higher plants, and animals, so no real new compound family was discovered but new derivatives. What was discovered is, onoceroids in fungi.

RESPONSE: We agree with the reviewer that the natural products obtained in our study lack structural novelty and understand that it is still challenging to tell which BGCs yield natural

products with a novel scaffold. However, we would like to emphasize that the biosynthetic mechanisms of fungal onocerooids are distinct from those of other organisms and that our approach indeed allows for the identification of novel biosynthetic mechanisms. As many Pyr4 homologues have already been identified and characterized, the discovery of a Pyr4 homologue with a new function would be challenging using a conventional approach (e.g., BLAST search to find Pyr4 homologues). Collectively, our study provides an alternative approach to accessing new enzymatic reactions, and therefore, the lack of structural novelty would not greatly influence the significance of our study.

3. The authors should provide at least some data on potential bioactivity of the fungal onocerooid derivatives.

RESPONSE: Actually, we have evaluated the antibacterial and antiviral activities of some of the compounds obtained in this study; however, no activity could be detected. In addition, some other compounds were difficult to dissolve in DMSO, which hampered the evaluation of their biological activities. Since the bioactivity of the obtained compounds is not the major focus of this study, we did not mention this in the manuscript. We would appreciate the reviewer's understanding on this matter.

4. Also, the authors state that „fungal onocerooid BGCs might be silent under standard laboratory conditions, which could explain why onocerooids had never been isolated from fungi.“ To lend weight to their discovery strategy it would be reasonable to measure the expression of a respective BGC at least in one of the identified fungi by RT-PCR or Northern blot.

RESPONSE: Regarding the metabolite analysis of the native strains, we have reconsidered the cultivation and analysis conditions, and we now found that *A. homomorphus* and *A. alliaceus* produce the onocerooids obtained from the *A. oryzae* transformant when they are cultivated in yeast extract sucrose (YES) broth and dextrin polypeptone yeast extract (DPY) broth, respectively. We also confirmed the expression of the onocerooid biosynthetic genes in these fungi by RT-PCR. We have revised the manuscript accordingly and added figures for their metabolite analysis and gene expression analysis in the Supplementary Information (Supplementary Figs. S5 and S6).

For *A. fumigatus*, although we have tried several solid and liquid media for cultivation, we were unable to confidently confirm the production of compound 7. Nevertheless, a very small peak that appears to be 7 was detected when the fungus was cultivated on a potato dextrose agar (PDA) plate (due to the technical difficulties in extracting RNA from the fungus cultivated on an agar plate, the gene expression analysis was not performed). The result of the metabolite analysis has been added to the Supplementary Information section (Supplementary Fig. S5).

Minor comments

1. Line 30: Please also cite Bergmann et al. 2007, Nat Chem Biol, who published the first example of expression of a silent gene cluster in fungi.

RESPONSE: This article has been cited as requested.

2. Line 50: "...selective extraction..." Please clarify, what do you mean? Selective extraction of BGCs for a distinct purpose?

RESPONSE: To clarify the point, we have changed "selective extraction of BGCs" to "selective extraction of BGCs with a desired or specific feature."

3. line 116: a non-reference strain is misleading, there is no accepted reference strain. Maybe you meant a non-sequenced strain?

RESPONSE: What we meant here was that the genome sequence of *Aspergillus fumigatus* A1163 is not a reference genome (as defined by the NCBI). We now understand that "nonreference strain" is misleading, and therefore, "*Aspergillus fumigatus* A1163, a nonreference strain of *A. fumigatus*, harbored a similar BGC" has been changed to "the genome of *Aspergillus fumigatus* A1163, a nonreference genome of *A. fumigatus*, contained a similar BGC."

4. line 126: Any idea why "most of the Pyr4-like proteins identified (1010/1050) are of ascomycete origin"?

RESPONSE: Perhaps Pyr4-family terpene cyclases have highly evolved among ascomycete fungi; however, unfortunately, we do not have any data to further discuss this observation at the moment. We appreciate the reviewers' understanding on this matter.

5. Please provide some more data on the recombinant strains of *Aspergillus*. Did you check how many copies of the plasmids were integrated into the genome? How did you proof that the strains contained the full plasmid?

RESPONSE: First of all, we would like to clarify the transformation method used in this study. The transformation of *Aspergillus oryzae* was performed by homologous recombination; each transformation plasmid was integrated onto a specific site of a chromosome of *A. oryzae*. Thus, the resultant transformants do not contain intact plasmids. Regarding the copy number, although we did not directly check it (and it would not be easy to do so), it is expected that there should be one copy of each gene due to the nature of homologous recombination. In addition, we would like to mention that the copy number is not critical for our work, as we do not aim for a quantitative analysis. We confirm the successful construction of each transformant by PCR-amplifying each

gene from the genomic DNA of the created transformants. We have now added a figure for the PCR result in the Supplementary Information (see Supplementary Fig. S11).

Reviewer #3

Dr. Jia Tang and Dr. Yudai Matsuda's study, titled 'Discovery of fungal onoceroid triterpenoids through domainless enzyme-targeted global genome mining,' introduces a new genome mining pipeline termed FunBGCeX (Fungal BGC eXtractor). The authors have performed heterologous expression experiments in *Aspergillus oryzae* using three homologous *Aspergillus* BGCs identified by FunBGCeX. Subsequent chemical validation of the resultant compounds was performed using GC-MS, HPLC, HRMS, NMR, and X-ray crystallography. Commendably, the authors have provided open access to the data generated by their FunBGCeX program through a well-documented GitHub repository, <https://github.com/ydmatsd/funbgcex>. In addition, they have established a user-friendly GUI website, accessible at <http://staffweb1.cityu.edu.hk/ymatsuda/funbgcs/funbgcs.html>, which is particularly beneficial for users with limited computational skills. The creation of this GUI website is a commendable effort to enhance the program's accessibility. Exploration of the interface revealed it to be highly intuitive, with useful several features, such as the integrated BLASTp functionality and the capability to copy the amino acid sequences of any protein. However, as detailed below, it is questionable if this works adds to the field of genome mining.

RESPONSE: We greatly appreciate the reviewer's time and a number of detailed comments and suggestions on our manuscript. We have revised our manuscript according to the reviewer's comments, and we believe that our manuscript has been greatly improved after the revision. Please see below our responses to each comment/question.

Comment 1 (Genome Mining): The authors present two major limitations in the leading fungal genome mining program:

Lines 40-41: "First, antiSMASH classifies many genes as "other genes," even when they encode a biosynthetic enzyme."

Lines 49-52: "Furthermore, although antiSMASH and the other aforementioned tools can be used to extract all possible BGCs from a genome sequence, they do not facilitate the selective extraction of BGCs. Thus, users need to select BGCs based on the criteria set for subsequent wet lab experiments."

As I understand their critiques, both could have been solved from a post-processing program/software as they revolve around how the antiSMASH predictions are presented to the

user (e.g., domains, category of gene, ability to filter), and not the underlying predictions themselves. As an example, one could have made a program that parses the output of antiSMASH and add in additional labels for mislabeled “other genes” while adding in options for the users to easily filter the antiSMASH BGC predictions by its presence of certain homologous genes, domains, or backbone classifications.

Given these two presented critiques, I struggle to see the need for an entirely new genome mining platform. The authors must better clarify what separates their new genome mining program from existing options? Additionally, as the authors opted to create an entirely new genome mining program it is important for them to sufficiently indicate how their program performs better or differently from existing genome mining alternative. On this point, I will indicate several lines of concern that I have:

RESPONSE: We agree with the reviewer that antiSMASH (or other existing tools) can be used to extract BGCs that meet one’s criteria. However, there are some limitations to antiSMASH-based analysis. First, if the method suggested by the reviewer is applied, all the BGCs must first be detected, after which BGCs with a specific feature will be extracted. This process will require a much longer time and is not feasible when a large number of genome sequences need to be analyzed. In addition, antiSMASH failed to detect some of the BGCs given in Fig. 2B, probably due to the lack of a well-known type of core enzyme gene.

To further demonstrate the utility of our genome mining method, we have heterologously expressed one of the BGCs that could not be detected by antiSMASH and obtained a new sesquiterpene hydroquinone named moserinol. Although moserinol lacks a structural novelty, our study has provided a new form of biosynthetic mechanism with a Pyr4 homologue. The manuscript has now been updated with these additional experimental results. Please note that we did not investigate the biosynthetic pathway of moserinol, as it is not the main focus of our study.

On lines 56-75 the authors created manually curated HMMER models for “domainless” proteins (as defined by it not having a detectable domain in the pFam database). From my rough estimation of Table S1 and S2 this led to the addition of between 50-100 previously unlabeled domains. The authors then created a set of rules based on the presence of biosynthetic domains labeled from their curated training database of already characterized clusters to determine the window size of these BGC predictions (line 76-88). It is important to note that while their addition of these 50-100 unlabeled domains is novel, the idea of using author specified rules to determine BGC boundaries is analogous to how many existing fungal genome mining programs have been created (including antiSMASH, SMURF).

RESPONSE: We agree that our methodology for determining BGC boundaries is not novel. To clarify this point, we have added the following phrase in this paragraph: “by applying a similar strategy used for other fungal genome mining tools, such as antiSMASH and SMURF.”

The authors present 1,050 BGCs that contained a homologous copy of the Pyr4 gene (lines 100-102). However, the authors fail to ever present a comparison between their algorithm and antiSMASH or other competitors. How many of these BGCs would have similarly been identified by antiSMASH, even if the Pyr4 gene is labeled as “Other Gene”? As the authors have not presented such data, it is difficult to make the case for using their program as opposed to existing solutions within the natural products community.

RESPONSE: To compare our method and antiSMASH, we have analyzed the same set of ~2,000 fungal genomes using antiSMASH and added one paragraph in the Results section to describe the analysis result (lines 147–158). We have also updated Figure 2B and Supplementary Data 3 to indicate which BGCs can (or cannot) be detected by antiSMASH. As mentioned above and described in the main text, we realized that there are some limitations to the antiSMASH-based analysis. Thus, we believe that our approach is advantageous when one would like to focus on BGCs with a specific feature.

The authors did validate three different Pyr4 containing proteins using heterologous expression (the chemistry and molecular biology in this paper will be addressed later in the review). However, as they state on (lines 209-211) “The onoceroid BGCs characterized in this study could also be identified using antiSMASH; ...” They counter this by stating that antiSMASH included more genes in the predictions than FunBGCeX (lines 210-211), however failed to present any data suggesting that the additional genes included by antiSMASH aren’t a part of the native BGC. Were these additional antiSMASH genes heterologously expressed? If they were not, what evidence do the authors have to suggest they are not involved in the production of the natural product in the native organism?

RESPONSE: Although we do not have direct evidence that determines the boundaries of the BGCs, we adopted a standard approach to predict the BGC boundaries (i.e., the nature of each gene in a genomic region and comparison with a similar genomic region in other fungi). For example, to predict the boundary of the *alli* cluster, we compared a genomic region containing the *alli* cluster with that from another fungus *Aspergillus leporis* CBS 151.66 and found that the five genes are highly conserved between the two fungal strains, whereas those in the flanking regions are not (see the figure below; the figure shows the blastp comparison of each gene product).

In addition, we understand that the precise prediction of BGC boundaries is still highly challenging, and our method often includes additional genes in extracted BGCs. We are not claiming that our method can more accurately predict the BGC boundaries. As mentioned above, the advantages of our approach are the shorter analysis time and the ability to extract BGCs that cannot be detected by antiSMASH.

The authors own admission that their validated BGCs could have been identified by the antiSMASH, in which the paper presents itself as a competitor, is concerning. If FunBGCeX is unable to prove that it can predict novel or different BGCs than antiSMASH I doubt it will have large adoption by the fungal natural product community. It also stands in direct contradiction with the line 215 where they state “the onoceroid BGCs would not have been effectively discovered using antiSMASH or conventional genome mining methodologies.” As the onoceroid BGC is detectable by antiSMASH, I fail to see how this statement is true.

RESPONSE: As already mentioned above, some of the BGCs detected in this study are not detectable by antiSMASH, and we have now added the characterization of such a BGC in the revised manuscript.

Regarding the second point, the antiSMASH-based approach will be more time-consuming and require an additional step to extract BGCs that satisfy the user’s requirement, as also discussed above. On the contrary, our method allows for the automated extraction of BGCs in a short period of time, which we believe is the advantage of our genome mining tool.

To summarize my concerns, I feel the authors have not shown any data to suggest their program performs any better than existing genome mining algorithms. Their validated BGCs could have been identified by antiSMASH and they utilized the same underlying framework that early iterations of antiSMASH used (i.e., rule based BGC calling). In addition, as the authors used a database of only characterized BGCs, their program falls under the same issue as many existing genome mining algorithms to being biased to only finding BGC architectures like what is already characterized.

RESPONSE: We would like to express our sincere appreciation for the reviewer's critiques and constructive suggestions on our bioinformatic analysis. Thanks to the comments, we now believe that our manuscript has been improved with some additional data. Although we are still unable to detect BGCs for something novel, as unpublished work, we have made some improvements in the bioinformatic tool, which can extract BGCs that lack a well-known type of core gene. We hope to publish another work with the improved genome mining method in the future.

Comment 2) Chemical analysis

The authors performed the structural analysis of eleven compounds including several new derivatives of onoceroid. Isolation, purification, and structure determination and elucidation of compounds were described well. I have some curiosities on the discussion and may ask that requires more details on the chemical analysis.

1. From Figure 3b, there is a small peak presented on the spectrum of NSAR1 extract, which is a control of this study. The peak disappeared on some of the mutants and also increased on others. The heterologous expressed genes may affect the chemical composition of the original primary/secondary metabolites of *A. oryzae*. In this case, I would recommend the authors to provide squalene and/or oxidosqualene production differences between control strain and mutants through LC-MS or GC-MS since the proposed biosynthetic pathway of new compounds shared the precursor squalene from the host stain. Squalene can be used to produce ergosterol or lanosterol.

RESPONSE: Although it is unclear what this small peak is, based on our experiences, the production levels of endogenous metabolites often vary due to uncertain reasons. Thus, it is likely that the difference in the peak size is caused by a reason other than the heterologous enzymes' squalene consumption. We also checked if we could detect squalene/oxidosqualene in the extracted metabolites; however, we were unable to detect these compounds. We would like to emphasize that the metabolite profiles of the transformants are consistent with the introduced genes, and therefore, the production level of the endogenous metabolite would not affect the conclusion of our manuscript. We would appreciate the reviewer's understanding on this matter.

2. From Figure 3c, it would be better to indicate new compounds with different colored numbers. For example, write numbers of known compounds in black while write numbers of new compounds in blue. It may be easier to recognize the similarities and differences between known and new compounds for the authors.

RESPONSE: As suggested, we have changed the color of the numbers for new compounds and modified the figure legend accordingly.

3. From the discussion section (line 236), the authors mentioned that “onoceroids have never been isolated from fungi. Thus, the present study provides the first examples of fungal onoceroids and their biosynthetic pathways.” Yes, onoceroids have never been isolated from fungi; however, onocerane carbon backbone was detected and isolated from fungi. Although this is first time the author isolated the onoceroids from heterologous expressed fungal genes, it would be hard to agree that this is the first examples of true fungal onoceroids since the compounds were not detected from the original fungal strains. It may be true that the compounds were derived from the fungal genome, but they are not natural products yet until they are isolated or detected from genetically unmodified fungal strains.

RESPONSE: As also mentioned in response to Reviewer 2, we have reconsidered the cultivation and analysis conditions and finally detected the onoceroid species in the three fungal strains (although the production of compound 7 could not be confidently confirmed).

4. From scheme 1, intermediate compounds 12 to 17 were not detected or identified. In this figure and the last paragraph of the discussion, it sounds like that they are the actual compounds. I would recommend removing the intermediates or clearly mentioned them they are theoretical intermediate compounds on the writing. Also, square brackets are recommended for the intermediates of biosynthetic pathway from the scheme 1.

RESPONSE: We appreciate the reviewer pointing this out. First, we mention that “The biosynthetic pathways of the onoceroids discovered in this study can be proposed as follows” in the first sentence of the paragraph to indicate that what is described in the paragraph is a proposed pathway. Since most of 12–17 are carbocationic intermediates, we feel that it is not confusing that these cationic species are actually isolated. We have mentioned that the other two, namely 15 and 18, have not been obtained in this study. Finally, as suggested, we have revised Scheme 1, in which biosynthetic/reaction intermediates are given in brackets.

5. For the new compounds, UVmax and FT-IR data are necessary.

RESPONSE: As requested, we have obtained UV and FT-IR spectra of the compounds obtained in this study and updated the analytical data part.

6. For the optical rotation, why is it recorded on different conditions? Usually, people use 23 or 20. The temperature should be the same. According to the references that the authors made, they were recorded under 23.

RESPONSE: Since our polarimeter does not have temperature control, it is difficult to perform all the measurements at the same temperature, even though the polarimeter is placed in a room

equipped with an air conditioner. As far as we understand, we need to report the temperature for a measurement, but there is no specific requirement to use a specific temperature.

7. Please provide combined ¹H table and ¹³C table for the compounds with the positions. In this case, it would be better to detect the differences easily (especially for the carbons). I understand that the authors already provided the separated tables for individual compounds from the supplementary information. However, it is still recommended.

RESPONSE: As requested, we have added combined NMR tables in the Supplementary Information (Supplementary Tables S10 and S11).

8. For the new compounds with overlaps such as compound 4, enlarged 2-D NMR of the overlapped section with correct indication is required. Otherwise, the authors may provide the raw NMR files for the supplementary data.

RESPONSE: We have uploaded all the NMR data to NP-MRD (NMR database for natural products; <https://np-mrd.org>) and will make the data public upon acceptance of this manuscript. Currently, the data can be accessed via this link: <https://depositions.np-mrd.org/request-data/44839824-8ed0-4837-89f6-4148b8c47a21>

9. For compound 4, is there a specific reason to take NMR with C₆D₆ instead of CDCl₃? As the authors already know, the NMR of most of the known onoceroids were identified and determined with CDCl₃. If there is no specific reason, it would be better to change the NMR data for compound 4 to CDCl₃ one.

RESPONSE: We normally use CDCl₃ as an NMR solvent as a first choice; however, for some compounds, the NMR data obtained in CDCl₃ do not allow for the complete structural determination, or NMR spectra in another solvent exhibit better resolution. For the cases where C₆D₆ was used, we found that C₆D₆ is a better solvent than CDCl₃ in terms of structural elucidation.

Comment 3) Biosynthetic pathway

1. While the onocenoids have prior not been detected in Fungi, the onocerane backbone, which is the same carbon backbone of onocenoids and upper chemical class of the compounds, have been characterized and found across many fungal taxa. At the end of the introduction (line 51), the authors emphasized obtaining a new class of metabolites through their genome mining tool. However, as there are several known biosynthetic pathways of the onocerane backbone in bacteria and plants, onocenoids cannot be correctly classified as a new class of natural products. To claim a new class of metabolites, the compounds should have either different carbon backbone or different backbone attachment instead of adding a single hydroxyl group or an acetyl group.

RESPONSE: We are afraid that we misunderstand the reviewer's point, but as far as we searched, we were unable to find fungal triterpenoids with the onocerane skeleton. In addition, review articles on fungal terpenoids and their biosynthesis (e.g., *Nat. Prod. Rep.*, 2014, **31**, 1449-1473; *Chin. J. Org. Chem.*, 2018, **38**, 2335-2347; *Nat. Prod. Rep.*, 2024, DOI: 10.1039/D3NP00052D) do not include onocerane triterpenoids from fungi. At least, we believe that the biosynthetic mechanism of fungal onocerane terpenoids has never been reported.

Nevertheless, it is true that onoceroids have been obtained from bacteria and plants and that the compounds obtained in this study lack a structural novelty. Therefore, we do not mention that we have obtained compounds with a new scaffold. However, one of the major points of this study would be the discovery of a biosynthetic mechanism that deviates from well-known ones, which could advance our understanding of natural product biosynthesis. In addition, although structurally not novel, we have indeed obtained several previously unreported compounds using our approach. Overall, along with the discovery of a new sesquiterpene hydroquinone, we believe that we have demonstrated the utility of our genome mining method in obtaining new organic molecules.

2. From line 58, the authors mentioned that "these domainless biosynthetic enzymes often drive diverse chemical reactions." A strong reference to support this claim must be included here. Some examples of domainless enzymes serving biosynthetic roles is not enough evidence that they are "often" biosynthetic as opposed to being a gap gene, pseudo gene, or having some alternative function. I would recommend removing the word "often".

RESPONSE: We have changed "often" to "occasionally" and "diverse" to "intriguing" in the sentence.

3. From line 105, a PaxC-like prenyltransferase is specifically required for the biosynthesis of indole sesquiterpenoids or diterpenoids. The given reference 29, only showed the example of indole diterpenoids. The authors may add and edit correct references.

RESPONSE: Actually, the reference describes the biosynthesis of spendole, which is an indole sesquiterpenoid.

4. From scheme 1, is it also possible that oxidosqualene is produced first and converted to the final products? When I read provided references of the biosynthetic pathways from bacteria and plants, it may also be possible to be converted from the oxidosqualene.

RESPONSE: We believe that it is unlikely that these compounds in Scheme 1 are synthesized from oxidosqualene, as the removal of the C-3 hydroxy group, which results from the epoxide

ring opening of oxidosqualene, requires additional biosynthetic steps, and enzymes for such reactions are not encoded by the onocerooid BGCs.

5. The authors should provide the detailed methods and results on the strain construction. To describe the biological results of the strain construction the following questions should be answered thoroughly:

1. Please provide the raw data and methodology showing the validation of the fungal transformation plasmids.

RESPONSE: We check the successful construction of the transformant by PCR-amplifying the introduced genes. We have added a figure for the result of the diagnostic PCR in the Supplementary Information (Supplementary Fig. S11).

2. Was the expression of the genes on the fungal transformation plasmids in *Aspergillus oryzae* validated (qPCR, semiQ, etc.). Please provide the data if so.

RESPONSE: We have checked the gene expression in selected transformants (*A. oryzae* with *homoS+M+B*, *fumiSI+M+B+P*, or *alliS+M+B+P+A*) by RT-PCR and added a figure in the Supplementary Information (Supplementary Fig. S12). In addition, we would like to mention that the metabolic profiles of the obtained transformants are consistent with the introduced genes, further providing evidence for the successful construction of the transformants and their gene expressions.

6. It appears that Figure 1S is nearly identical to Figure 1 in Matsuda's 2018 Nature Communications. To avoid any issues with self-plagiarism this figure should be re-drawn.

RESPONSE: Actually, Figure S1A was newly created for this manuscript, and we did not copy it from our previous paper. We agree that these two figures are somewhat similar (as they are both for the same BGC), but we believe that they are sufficiently different (see below for the comparison of the two figures).

Figure S1A of this manuscript.

Fig. 1b from *Nat. Commun.* 2018, 9, 2587.

Reviewer #4

Matsuda et al reported the discovery of a series of onoceroid compounds encoded by fungal biosynthetic gene clusters (BGCs). Those compounds were not isolated from the producing strain but generated through in vivo co-expression of different combinations of biosynthetic genes from the putative BGCs. A rigorous structural characterization was carried out by NMR, MS, Mosher's method and X-ray crystallography. Authors also described in detail a genome mining tool developed for fungal BGCs that incorporates a manually curated in-house database. Using this tool, authors input Pyr4-family terpene cyclases as the target gene, and found a number of BGCs that may encode novel terpene structures. Although this work is interesting, the overall significance of the genome mining method, and the significance of the function of Pyr4-family terpene cyclase is in question. There are several aspects mentioned below as suggestions to improve this manuscript. If significant improvements cannot be made, it is suggested that this work is too preliminary for *Nat. Comm.* and may be more suitable for more specialized journals.

RESPONSE: We would like to express our sincere appreciation to the reviewer for providing invaluable comments and suggestions on our manuscript. Thanks to all the reviewers' comments, we believe that our manuscript has been significantly improved after the revision. We hope that the reviewer will find it suitable for publication. Please see below our responses to each comment/question.

1. Development of a fungal genome mining tool

Since this tool contains a manually curated database, it is expected to be more accurate in gene annotation for natural product biosynthetic enzymes. Except that, it seems that current genome mining tools (such as antiSMASH) can achieve all the functions claimed for this tool. In addition, there seems to be a lot of manual manipulation of data when using this tool, which does not sound user-friendly comparing to antiSMASH. In this manuscript, authors only used this tool to discover Pyr4-like protein containing BGCs. Perhaps authors could consider discover more types of BGCs using different proteins to show the broad applicability of this tool?

RESPONSE: We have used antiSMASH many times and understand that it has many sophisticated and useful functions that are not present in ours. Actually, antiSMASH can also detect the onocerooid BGCs studied in this manuscript. However, as also mentioned in the revised manuscript (lines 147–158), the discovery of these BGCs would require somewhat tedious processes with antiSMASH: (1) detection of all BGCs from fungal genomes by antiSMASH; (2) extraction of all protein sequences from the detected BGCs; (3) BLAST (or equivalent) search to identify Pyr4 homologues; and (4) retrieval of the BGCs of interest. Among these four steps, steps 2–4 need to be manually done by the user. In contrast, our methodology can automatically extract BGCs with *pyr4* homologues (or any other types of proteins of user's interest) from a given fungal genome, which we believe is an advantageous point of our genome mining method.

To further demonstrate the utility of our method, we heterologously expressed one of the BGCs that are not detected by antiSMASH and obtained a new sesquiterpene hydroquinone named moserinol, providing a new form of biosynthetic mechanism with a Pyr4-like terpene cyclase. We have now updated the manuscript with this additional experimental data.

In terms of the genome mining targeting enzymes other than Pyr4 homologues, although we do not have any experimental data at the moment, as shown in the Discussion part, we have used our tool to mine BGCs with a *pydY* homologue and discovered some BGCs with interesting features. We believe that this bioinformatic analysis suggests the broad applicability of our method.

2. Function of Pyr4-family terpene cyclase

In this manuscript, authors only used heterologous host to express different combinations of biosynthetic proteins, identified corresponding products, and used those data to predict the biosynthetic path and claimed the terpene cyclase-catalyzed reaction. To more rigorously characterize this novel cyclase, the enzyme should be studied in more detail. The true substrate and any possible cofactors need to be incubated with the enzyme in vitro, and product structures need to be characterized. Mechanistic studies by mutagenesis and co-crystallization can also add more significance of this study.

RESPONSE: Pyr4-family terpene cyclases are transmembrane proteins and thus are difficult to be purified; although a number of Pyr4 homologues have already been identified, to the best of our knowledge, none of them has been successfully purified for in vitro enzymatic reactions or structural biology studies. Therefore, unfortunately, the suggested experiments would be difficult to perform. Meanwhile, we recently reported in-depth characterizations of some Pyr4-family terpene cyclases using the AlphaFold2-generated protein models (*Angew. Chem. Int. Ed.* 2023, **62**, e202306046), providing plausible reaction mechanisms for this class of terpene cyclases. We could apply a similar methodology to further investigate the functions of the three terpene cyclases; however, the enzymology of these cyclases is not the main focus of this manuscript, and therefore, we would like to perform further studies, such as mutational experiments and functional conversions, as another work. We would appreciate the reviewer's understanding on this matter.

3. Gene cluster identification and expression

The characterized structures are carbon skeleton with minimum oxidation. Authors also stated that compound 8 is a biosynthetic precursor of a natural product. Considering the fact that they are heterologously expressed using at most five proteins, is it possible that this type of BGC have greater potential for structural diversity? Wild-type strains containing this type of BGC can be isolated to find more structural analogs. In addition, Fig 2 panel b showed a couple of such BGCs contain "other" proteins. Heterologous expression of the whole cluster (not just selected proteins) with "other" genes can also be attempted.

RESPONSE: We agree with the reviewer that further characterization of other BGCs could lead to the discovery of additional onoceroid species. However, we have already characterized three different BGCs in the present study, and the characterization of some more onoceroid BGCs would not be of great interest. Instead, as also mentioned above, we now report the characterization of another type of BGC, which is not detectable by antiSMASH. We believe that this additional experiment has improved the quality of our manuscript.

Regarding the "other" genes, these genes encode a protein that is apparently not a biosynthetic protein or a truncated biosynthetic protein. Thus, we reasoned that these genes are not involved in biosynthetic processes.

4. Why did authors conduct research on fungal onoceroid triterpenoids

If the authors can explore interesting medical uses or any other applications of those products, the novelty of this manuscript can be improved.

RESPONSE: Although we investigated the biological activities of the obtained onoceroids, unfortunately, we have not found any activities from these compounds so far. Nevertheless, the

biological activity or application of the obtained compounds is not the main focus of our work, and we believe that our manuscript provides a useful method for fungal genome mining.

REVIEWERS' COMMENTS

Reviewer #2 (Remarks to the Author):

The authors fully addressed my major concerns

(1) They now provide evidence that the mining tool they developed has some advantages compared to antiSMASH because they added data on the heterologous expressing of one of the BGCs that were not detected by antiSMASH. As a results, the authors obtained a new sesquiterpene hydroquinone named moserinol. Although it is not really a novel structure, I believe this proof of concept is convincing.

(2) The authors confirmed the expression of the onoceroid biosynthetic genes in these fungi by RT-PCR. Although the authors now realized for some clusters that they are apparently not silent. The data were added (Supplementary Figs. S5, S6, S12).

(3) The genetic characterization is not really state-of-the art but the strains serve the purpose. My other points did not need further experiments.

Reviewer #3 (Remarks to the Author):

The authors did a commendable job in responding to the critiques. I have nothing further to add.

Reviewer #3 (Remarks on code availability):

I don't even know what you mean by the code so I said no.

Reviewer #4 (Remarks to the Author):

Given the relatively low novelty and limited application of the tool developed in this article, I still think that this work is too preliminary for Nat. Comm., and may be more suitable for other journals.